# Regularization Matters: Generalization and Optimization of Neural Nets v.s. their Induced Kernel

**Colin Wei**
Department of Computer Science
Stanford University
colinwei@stanford.edu

**Jason D. Lee**
Department of Electrical Engineering
Princeton University
jasonlee@princeton.edu

**Qiang Liu**
Department of Computer Science
University of Texas at Austin
lqiang@cs.texas.edu

**Tengyu Ma**
Department of Computer Science
Stanford University
tengyuma@stanford.edu

## Abstract

Recent works have shown that on sufficiently over-parametrized neural nets, gradient descent with relatively large initialization optimizes a prediction function in the RKHS of the Neural Tangent Kernel (NTK). This analysis leads to global convergence results but does not work when there is a standard $\ell_2$ regularizer, which is useful to have in practice. We show that sample efficiency can indeed depend on the presence of the regularizer: we construct a simple distribution in $d$ dimensions which the optimal regularized neural net learns with $O(d)$ samples but the NTK requires $\Omega(d^2)$ samples to learn. To prove this, we establish two analysis tools: i) for multi-layer feedforward ReLU nets, we show that the global minimizer of a weakly-regularized cross-entropy loss is the max normalized margin solution among all neural nets, which generalizes well; ii) we develop a new technique for proving lower bounds for kernel methods, which relies on showing that the kernel cannot focus on informative features. Motivated by our generalization results, we study whether the regularized global optimum is attainable. We prove that for infinite-width two-layer nets, noisy gradient descent optimizes the regularized neural net loss to a global minimum in polynomial iterations.

## 1 Introduction

In deep learning, over-parametrization refers to the widely-adopted technique of using more parameters than necessary [35, 40]. Over-parametrization is crucial for successful optimization, and a large body of work has been devoted towards understanding why. One line of recent works [17, 37, 22, 21, 2, 76, 31, 6, 16, 72] offers an explanation that invites analogy with kernel methods, proving that with sufficient over-parameterization and a certain initialization scale and learning rate schedule, gradient descent essentially learns a linear classifier on top of the initial random features. For this same setting, Daniely [17], Du et al. [22, 21], Jacot et al. [31], Arora et al. [6, 5] make this connection explicit by establishing that the prediction function found by gradient descent is in the span of the training data in a reproducing kernel Hilbert space (RKHS) induced by the Neural Tangent Kernel (NTK). The generalization error of the resulting network can be analyzed via the Rademacher complexity of the kernel method.

These works provide some of the first algorithmic results for the success of gradient descent in optimizing neural nets; however, the resulting generalization error is only as good as that of fixed

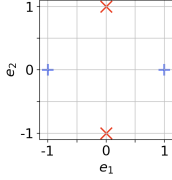

Figure 1: Datapoints from $\mathcal{D}$ have first two coordinates displayed above, with red and blue denoting labels of -1, +1, respectively. The remaining coordinates are uniform in $\{-1, +1\}^{d-2}$.

kernels [6]. On the other hand, the equivalence of gradient descent and NTK is broken if the loss has an explicit regularizer such as weight decay.

In this paper, we study the effect of an explicit regularizer on neural net generalization via the lens of margin theory. We first construct a simple distribution on which the two-layer network optimizing explicitly regularized logistic loss will achieve a large margin, and therefore, good generalization. On the other hand, any prediction function in the span of the training data in the RKHS induced by the NTK will overfit to noise and therefore achieve poor margin and bad generalization.

**Theorem 1.1** (Informal version of Theorem 2.1). *Consider the setting of learning the distribution $\mathcal{D}$ defined in Figure 1 using a two-layer network with relu activations with the goal of achieving small generalization error. Using $o(d^2)$ samples, no function in the span of the training data in the RKHS induced by the NTK can succeed. On the other hand, the global optimizer of the $\ell_2$-regularized logistic loss can learn $\mathcal{D}$ with $O(d)$ samples.*

The full result is stated in Section 2. The intuition is that regularization allows the neural net to obtain a better margin than the fixed NTK kernel and thus achieve better generalization. Our sample complexity lower bound for NTK applies to a broad class of losses including standard 0-1 classification loss and squared $\ell_2$. To the best of our knowledge, the proof techniques for obtaining this bound are novel and of independent interest (see our proof overview in Section 2). In Section 5, we confirm empirically that an explicit regularizer can indeed improve the margin and generalization.

Yehudai and Shamir [73] also prove a lower bound on the learnability of neural net kernels. They show that an approximation result that $\Omega(\exp(d))$ random relu features are required to fit a single neuron in $\ell_2$ squared loss, which lower bounds the amount of over-parametrization necessary to approximate a single neuron. In contrast, we prove sample-complexity lower bounds which hold for both classification and $\ell_2$ loss even with *infinite over-parametrization*.

Motivated by the provably better generalization of regularized neural nets for our constructed instance, in Section 3 we study their optimization, as the previously cited results only apply when the neural net behaves like a kernel. We show optimization is possible for infinite-width regularized nets.

**Theorem 1.2** (Informal, see Theorem 3.3). *For infinite-width two layer networks with $\ell_2$-regularized loss, noisy gradient descent finds a global optimizer in a polynomial number of iterations.*

This improves upon prior works [43, 15, 65, 61] which study optimization in the same infinite-width limit but do not provide polynomial convergence rates. (See more discussions in Section 3.)

To establish Theorem 1.1, we rely on tools from margin theory. In Section 4, we prove a number of results of independent interest regarding the margin of a regularized neural net. We show that the global minimum of weakly-regularized logistic loss of any homogeneous network (regardless of depth or width) achieves the max normalized margin among all networks with the same architecture (Theorem 4.1). By "weak" regularizer, we mean that the coefficient of the regularizer in the loss is very small (approaching 0). By combining with a result of [25], we conclude that the minimizer enjoys a width-free generalization bound depending on only the inverse normalized margin (normalized by the norm of the weights) and depth (Corollary 4.2). This explains why optimizing the $\ell_2$-regularized loss typically used in practice can lead to parameters with a large margin and good generalization. We further note that the maximum possible margin is non-decreasing in the width of the architecture, so the generalization bound of Corollary 4.2 improves as the size of the network grows (see Theorem 4.3). Thus, even if the dataset is already separable, it could still be useful to further over-parameterize to achieve better generalization.

Finally, we empirically validate several claims made in this paper in Section 5. First, we confirm on synthetic data that neural networks do generalize better with an explicit regularizer vs. without. Second, we show that for two-layer networks, the test error decreases and margin increases as the hidden layer grows, as predicted by our theory.

## 1.1 Additional Related Work

Zhang et al. [74] and Neyshabur et al. [52] show that neural network generalization defies conventional explanations and requires new ones. Neyshabur et al. [48] initiate the search for the "inductive bias" of neural networks towards solutions with good generalization. Recent papers [30, 12, 14] study inductive bias through training time and sharpness of local minima. Neyshabur et al. [49] propose a steepest descent algorithm in a geometry invariant to weight rescaling and show this improves generalization. Morcos et al. [45] relate generalization to the number of "directions" in the neurons. Other papers [26, 68, 46, 28, 38, 27, 38, 32] study implicit regularization towards a specific solution. Ma et al. [41] show that implicit regularization helps gradient descent avoid overshooting optima. Rosset et al. [58, 59] study linear logistic regression with weak regularization and show convergence to the max margin. In Section 4, we adopt their techniques and extend their results.

A line of work initiated by Neyshabur et al. [50] has focused on deriving tighter norm-based Rademacher complexity bounds for deep neural networks [9, 51, 25] and new compression based generalization properties [4]. Bartlett et al. [9] highlight the important role of normalized margin in neural net generalization. Wei and Ma [70] prove generalization bounds depending on additional data-dependent properties. Dziugaite and Roy [23] compute non-vacuous generalization bounds from PAC-Bayes bounds. Neyshabur et al. [53] investigate the Rademacher complexity of two-layer networks and propose a bound that is decreasing with the distance to initialization. Liang and Rakhlin [39] and Belkin et al. [10] study the generalization of kernel methods.

For optimization, Soudry and Carmon [67] explain why over-parametrization can remove bad local minima. Safran and Shamir [63] show over-parametrization can improve the quality of a random initialization. Haeffele and Vidal [29], Nguyen and Hein [55], and Venturi et al. [69] show that for sufficiently overparametrized networks, all local minima are global, but do not show how to find these minima via gradient descent. Du and Lee [19] show for two-layer networks with quadratic activations, all second-order stationary points are global minimizers. Arora et al. [3] interpret over-parametrization as a means of acceleration. Mei et al. [43], Chizat and Bach [15], Sirignano and Spiliopoulos [65], Dou and Liang [18], Mei et al. [44] analyze a distributional view of over-parametrized networks. Chizat and Bach [15] show that Wasserstein gradient flow converges to global optimizers under structural assumptions. We extend this to a polynomial-time result.

Finally, many papers have shown convergence of gradient descent on neural nets [2, 1, 37, 22, 21, 6, 76, 13, 31, 16] using analyses which prove the weights do not move far from initialization. These analyses do not apply to the regularized loss, and our experiments in Section F suggest that moving away from the initialization is important for better test performance.

Another line of work takes a Bayesian perspective on neural nets. Under an appropriate choice of prior, they show an equivalence between the random neural net and Gaussian processes in the limit of infinite width or channels [47, 71, 36, 42, 24, 56]. This provides another kernel perspective of neural nets.

Yehudai and Shamir [73], Chizat and Bach [16] also argue that the kernel perspective of neural nets is not sufficient for understanding the success of deep learning. Chizat and Bach [16] argue that the kernel perspective of gradient descent is caused by a large initialization and does not necessarily explain the empirical successes of over-parametrization. Yehudai and Shamir [73] prove that $\Omega(\exp(d))$ random relu features cannot approximate a single neuron in squared error loss. In comparison, our lower bounds are for the sample complexity rather than width of the NTK prediction function and apply even with infinite over-parametrization for both classification and squared loss.

## 1.2 Notation

Let $\mathbb{R}$ denote the set of real numbers. We will use $\|\cdot\|$ to indicate a general norm, with $\|\cdot\|_2$ denoting the $\ell_2$ norm and $\|\cdot\|_F$ the Frobenius norm. We use $^-$ on top of a symbol to denote a unit vector: when applicable, $\bar{u} \triangleq u/\|u\|$, with the norm $\|\cdot\|$ clear from context. Let $\mathcal{N}(0, \sigma^2)$ denote the normal distribution with mean 0 and variance $\sigma^2$. For vectors $u_1 \in \mathbb{R}^{d_1}$, $u_2 \in \mathbb{R}^{d_2}$, we use the notation $(u_1, u_2) \in \mathbb{R}^{d_1+d_2}$ to denote their concatenation. We also say a function $f$ is $a$-homogeneous in input $x$ if $f(cx) = c^a f(x)$ for any $c$, and we say $f$ is $a$-positive-homogeneous if there is the additional constraint $c > 0$. We reserve the symbol $X = [x_1, \ldots, x_n]$ to denote the collection of datapoints (as a matrix), and $Y = [y_1, \ldots, y_n]$ to denote labels. We use $d$ to denote the dimension of our data.

We will use the notations $a \lesssim b$, $a \gtrsim b$ to denote less than or greater than up to a universal constant, respectively, and when used in a condition, to denote the existence of such a constant such that the condition is true. Unless stated otherwise, $O(\cdot), \Omega(\cdot)$ denote some universal constant in upper and lower bounds. The notation poly denotes a universal constant-degree polynomial in the arguments.

## 2  Generalization of Regularized Neural Net vs. NTK Kernel

We will compare neural net solutions found via regularization and methods involving the NTK and construct a data distribution $\mathcal{D}$ in $d$ dimensions which the neural net optimizer of regularized logistic loss learns with sample complexity $O(d)$. The kernel method will require $\Omega(d^2)$ samples to learn.

We start by describing the distribution $\mathcal{D}$ of examples $(x, y)$. Here $e_i$ is the i-th standard basis vector and we use $x^\top e_i$ to represent the $i$-coordinate of $x$ (since the subscript is reserved to index training examples). First, for any $k \geq 3, x^\top e_k \sim \{-1, +1\}$ is a uniform random bit, and for $x^\top e_1, x^\top e_2$ and $y$, choose

$$
\begin{aligned}
y = +1, & \quad x^\top e_1 = +1, & \quad x^\top e_2 = 0 & \quad \text{w/ prob. } 1/4 \\
y = +1, & \quad x^\top e_1 = -1, & \quad x^\top e_2 = 0 & \quad \text{w/ prob. } 1/4 \\
y = -1, & \quad x^\top e_1 = 0, & \quad x^\top e_2 = +1 & \quad \text{w/ prob. } 1/4 \\
y = -1, & \quad x^\top e_1 = 0, & \quad x^\top e_2 = -1 & \quad \text{w/ prob. } 1/4
\end{aligned}
\tag{2.1}
$$

The distribution $\mathcal{D}$ contains all of its signal in the first 2 coordinates, and the remaining $d - 2$ coordinates are noise. We visualize its first 2 coordinates in Figure 1.

Next, we formally define the two layer neural net with relu activations and its associated NTK. We parameterize a two-layer network with $m$ units by last layer weights $w_1, \ldots, w_m \in \mathbb{R}$ and weight vectors $u_1, \ldots, u_m \in \mathbb{R}^d$. We denote by $\Theta$ the collection of parameters and by $\theta_j$ the unit-$j$ parameters $(u_j, w_j)$. The network computes $f^{\text{NN}}(x; \Theta) \triangleq \sum_{j=1}^m w_j [u_j^\top x]_+$, where $[\cdot]_+$ denotes the relu activation. For binary labels $y_1, \ldots, y_n \in \{-1, +1\}$, the $\ell_2$ regularized logistic loss is

$$
L_\lambda(\Theta) \triangleq \frac{1}{n} \sum_{i=1}^n \log(1 + \exp(-y_i f^{\text{NN}}(x_i; \Theta))) + \lambda \|\Theta\|_F^2
\tag{2.2}
$$

Let $\Theta_\lambda \in \arg\min_\Theta L_\lambda(\Theta)$ be its global optimizer. Define the NTK kernel associated with the architecture (with random weights):

$$
K(x', x) = \mathbb{E}_{w \sim \mathcal{N}(0, r_w^2), u \sim \mathcal{N}(0, r_u^2 I)} \left[ \langle \nabla_\theta f^{\text{NN}}(x; \Theta), \nabla_\theta f^{\text{NN}}(x'; \Theta) \rangle \right]
$$

where $\nabla_\theta f^{\text{NN}}(x; \Theta) = (w \mathbb{1}(x^\top u \geq 0)x, [x^\top u]_+)$ is the gradient of the network output with respect to a generic hidden unit, and $r_w, r_u$ are relative scaling parameters. Note that the typical NTK is realized specifically with scales $r_w = r_u = 1$, but our bound applies for all choices of $r_w, r_u$.

For coefficients $\beta$, we can then define the prediction function $f^{\text{kernel}}(x; \beta)$ in the RKHS induced by $K$ as $f^{\text{kernel}}(x; \beta) \triangleq \sum_{i=1}^n \beta_i K(x_i, x)$. For example, such a classifier would be attained by running gradient descent on squared loss for a wide network using the appropriate random initialization (see [31, 22, 21, 6]). We now present our comparison theorem below and fill in its proof in Section B.

**Theorem 2.1.** *Let $\mathcal{D}$ be the distribution defined in equation 2.1. With probability $1 - d^{-5}$ over the random draw of $n \lesssim d^2$ samples $(x_1, y_1), \ldots, (x_n, y_n)$ from $\mathcal{D}$, for all choices of $\beta$, the kernel prediction function $f^{\text{kernel}}(\cdot; \beta)$ will have at least $\Omega(1)$ error:*

$$
\Pr_{(x,y) \sim \mathcal{D}}[f^{\text{kernel}}(x; \beta)y \leq 0] = \Omega(1)
$$

*Meanwhile, for $\lambda \leq \text{poly}(n)^{-1}$, the regularized neural net solution $f^{\text{NN}}(\cdot; \Theta_\lambda)$ with at least 4 hidden units can have good generalization with $O(d^2)$ samples because we have the following generalization error bound:*

$$
\Pr_{(x,y) \sim \mathcal{D}}[f^{\text{NN}}(x; \Theta_\lambda)y \leq 0] \lesssim \sqrt{\frac{d}{n}}
$$

*This implies a $\Omega(d)$ sample-complexity gap between the regularized neural net and kernel prediction function.*

While the above theorem is stated for classification, the same $\mathcal{D}$ can be used to straightforwardly prove a $\Omega(d)$ sample complexity gap for the truncated squared loss $\ell(\hat{y}; y) = \min((y - \hat{y})^2, 1)$.[1] We provide more details in Section B.3.

Our intuition of this gap is that the regularization allows the neural net to find informative features (weight vectors), that are adaptive to the data distribution and easier for the last layers' weights to separate. For example, the neurons $[e_1 x]_+, [-e_1 x]_+, [e_2 x]_+, [-e_2 x]_+$ are enough to fit our particular distribution. In comparison, the NTK method is unable to change the feature space and is only searching for the coefficients in the kernel space.

*Proof techniques for the upper bound:* For the upper bound, neural nets with small Euclidean norm will be able to separate $\mathcal{D}$ with large margin (a two-layer net with width 4 can already achieve a large margin). As we show in Section 4, a solution with a max neural-net margin is attained by the global optimizer of the regularized logistic loss — in fact, we show this holds for generally homogeneous networks of any depth and width (Theorem 4.1). Then, by the classical connection between margin and generalization [34], this optimizer will generalize well.

*Proof techniques for the lower bound:* On the other hand, the NTK will have a worse margin when fitting samples from $\mathcal{D}$ than the regularized neural networks because NTK operates in a fixed kernel space.[2] However, proving that the NTK has a small margin does not suffice because the generalization error bounds which depend on margin may not be tight.

We develop a new technique to prove lower bounds for kernel methods, which we believe is of independent interest, as there are few prior works that prove lower bounds for kernel methods. (One that does is [54], but their results require constructing an artificial kernel and data distribution, whereas our lower bounds are for a fixed kernel.) The main intuition is that because NTK uses infinitely many random features, it is difficult for the NTK to focus on a small number of informative features – doing so would require a very high RKHS norm. In fact, we show that with a limited number of examples, any function that in the span of the training examples must heavily use random features rather than informative features. The random features can collectively fit the training data, but will give worse generalization.

## 3 Perturbed Wasserstein Gradient Flow Finds Global Optimizers in Polynomial Time

In the prior section, we argued that a neural net with $\ell_2$ regularization can achieve much better generalization than the NTK. Our result required attaining the global minimum of the regularized loss; however, existing optimization theory only allows for such convergence to a global minimizer with a large initialization and no regularizer. Unfortunately, these are the regimes where the neural net learns a kernel prediction function [31, 22, 6].

In this section, we show that at least for infinite-width two-layer nets, optimization is not an issue: noisy gradient descent finds global optimizers of the $\ell_2$ regularized loss in polynomial iterations.

Prior work [43, 15] has shown that as the hidden layer size grows to infinity, gradient descent for a finite neural network approaches the Wasserstein gradient flow over distributions of hidden units (defined in equation 3.1). With the assumption that the gradient flow converges, which is non-trivial since the space of distributions is infinite-dimensional, Chizat and Bach [15] prove that Wasserstein gradient flow converges to a global optimizer but do not specify a rate. Mei et al. [43] add an entropy regularizer to form an objective that is the infinite-neuron limit of stochastic Langevin dynamics. They show global convergence but also do not provide explicit rates. In the worst case, their convergence can be exponential in dimension. In contrast, we provide explicit *polynomial* convergence rates for a slightly different algorithm, perturbed Wasserstein gradient flow.

Infinite-width neural nets are modeled mathematically as a distribution over weights: formally, we optimize the following functional over distributions $\rho$ on $\mathbb{R}^{d+1}$: $L[\rho] \triangleq R(\int \Phi d\rho) + \int V d\rho$, where $\Phi : \mathbb{R}^{d+1} \to \mathbb{R}^k$, $R : \mathbb{R}^k \to \mathbb{R}$, and $V : \mathbb{R}^{d+1} \to \mathbb{R}$. $R$ and $V$ can be thought of as the loss and regularizer, respectively. In this work, we consider 2-homogeneous $\Phi$ and $V$. We will additionally

require that $R$ is convex and nonnegative and $V$ is positive on the unit sphere. Finally, we need standard regularity assumptions on $R, \Phi$, and $V$:

**Assumption 3.1** (Regularity conditions on $\Phi$, $R$, $V$). *$\Phi$ and $V$ are differentiable as well as upper bounded and Lipschitz on the unit sphere. $R$ is Lipschitz and its Hessian has bounded operator norm.*

We provide more details on the specific parameters (for boundedness, Lipschitzness, etc.) in Section E.1. We note that relu networks satisfy every condition but differentiability of $\Phi$.[3] We can fit a $\ell_2$ regularized neural network under our framework:

**Example 3.2** (Logistic loss for neural networks). *We interpret $\rho$ as a distribution over the parameters of the network. Let $k \triangleq n$ and $\Phi_i(\theta) \triangleq w\phi(u^\top x_i)$ for $\theta = (w, u)$. In this case, $\int \Phi d\rho$ is a distributional neural network that computes an output for each of the $n$ training examples (like a standard neural network, it also computes a weighted sum over hidden units). We can compute the distributional version of the regularized logistic loss in equation 2.2 by setting $V(\theta) \triangleq \lambda\|\theta\|_2^2$ and $R(a_1, \dots, a_n) \triangleq \sum_{i=1}^n \log(1 + \exp(-y_i a_i))$.*

We will define $L'[\rho] : \mathbb{R}^{d+1} \to \mathbb{R}$ with $L'[\rho](\theta) \triangleq \langle R'(\int \Phi d\rho), \Phi(\theta)\rangle + V(\theta)$ and $v[\rho](\theta) \triangleq -\nabla_\theta L'[\rho](\theta)$. Informally, $L'[\rho]$ is the gradient of $L$ with respect to $\rho$, and $v$ is the induced velocity field. For the standard Wasserstein gradient flow dynamics, $\rho_t$ evolves according to

$$\frac{d}{dt}\rho_t = -\nabla \cdot (v[\rho_t]\rho_t) \tag{3.1}$$

where $\nabla\cdot$ denotes the divergence of a vector field. For neural networks, these dynamics formally define continuous-time gradient descent when the hidden layer has infinite size (see Theorem 2.6 of [15], for instance). More generally, equation 3.1 is due to the formula for Wasserstein gradient flow dynamics (see for example [64]), which are derived via continuous-time steepest descent with respect to Wasserstein distance over the space of probability distributions on the neurons. We propose the following modified dynamics:

$$\frac{d}{dt}\rho_t = -\sigma\rho_t + \sigma U^d - \nabla \cdot (v[\rho_t]\rho_t) \tag{3.2}$$

where $U^d$ is the uniform distribution on $\mathbb{S}^d$. In our perturbed dynamics, we add very small uniform noise over $U^d$, which ensures that at all time-steps, there is sufficient mass in a descent direction for the algorithm to decrease the objective. For infinite-size neural networks, one can informally interpret this as re-initializing a very small fraction of the neurons at every step of gradient descent. We prove convergence to a global optimizer in time polynomial in $1/\epsilon, d$, and the regularity parameters.

**Theorem 3.3** (Theorem E.4 with regularity parameters omitted). *Suppose that $\Phi$ and $V$ are 2-homogeneous and the regularity conditions of Assumption 3.1 are satisfied. Also assume that from starting distribution $\rho_0$, a solution to the dynamics in equation 3.2 exists. Define $L^\star \triangleq \inf_\rho L[\rho]$. Let $\epsilon > 0$ be a desired error threshold and choose $\sigma \triangleq \exp(-d\log(1/\epsilon)\mathrm{poly}(k, L[\rho_0] - L^\star))$ and $t_\epsilon \triangleq \frac{d^2}{\epsilon^4}\mathrm{poly}(\log(1/\epsilon), k, L[\rho_0] - L^\star)$, where the regularity parameters for $\Phi$, $V$, and $R$ are hidden in the $\mathrm{poly}(\cdot)$. Then, perturbed Wasserstein gradient flow converges to an $\epsilon$-approximate global minimum in $t_\epsilon$ time:*

$$\min_{0 \le t \le t_\epsilon} L[\rho_t] - L^\star \le \epsilon$$

We state and prove a version of Theorem 3.3 that includes regularity parameters in Sections E.1 and E.3. The key idea for the proof is as follows: as $R$ is convex, the optimization problem will be convex over the space of distributions $\rho$. This convexity allows us to argue that if $\rho$ is suboptimal, there either exists a descent direction $\bar{\theta} \in \mathbb{S}^d$ where $L'[\rho](\bar{\theta}) \ll 0$, or the gradient flow dynamics will result in a large decrease in the objective. If such a direction $\bar{\theta}$ exists, the uniform noise $\sigma U^d$ along with the 2-homogeneity of $\Phi$ and $V$ will allow the optimization dynamics to increase the mass in this direction exponentially fast, which causes a polynomial decrease in the loss.

As a technical detail, Theorem 3.3 requires that a solution to the dynamics exists. We can remove this assumption by analyzing a discrete-time version of equation 3.2: $\rho_{t+1} \triangleq \rho_t + \eta(-\sigma\rho_t + \sigma U^d - \nabla \cdot$

$(v[\rho_t]\rho_t))$, and additionally assuming $\Phi$ and $V$ have Lipschitz gradients. In this setting, a polynomial time convergence result also holds. We state the result in Section E.4.

An implication of our Theorem 3.3 is that for infinite networks, we can optimize the weakly-regularized logistic loss in time polynomial in the problem parameters and $\lambda^{-1}$. In Theorem 2.1 we only require $\lambda^{-1} = \text{poly}(n)$; thus, an infinite width neural net can learn the distribution $\mathcal{D}$ up to error $\tilde{O}(\sqrt{d/n})$ in polynomial time using noisy gradient descent.

## 4   Weak Regularizer Guarantees Max Margin Solutions

In this section, we collect a number of results regarding the margin of a regularized neural net. These results provide the tools for proving generalization of the weakly-regularized NN solution in Theorem 2.1. The key technique is showing that with small regularizer $\lambda \to 0$, the global optimizer of regularized logistic loss will obtain a maximum margin. It is well-understood that a large neural net margin implies good generalization performance [9].

In fact, our result applies to a function class much broader than two-layer relu nets: in Theorem 4.1 we show that when we add a weak regularizer to cross-entropy loss with *any* positive-homogeneous prediction function, the normalized margin of the optimum converges to the max margin. For example, Theorem 4.1 applies to feedforward relu networks of arbitrary depth and width. In Theorem C.2, we bound the approximation error in the maximum margin when we only obtain an approximate optimizer of the regularized loss. In Corollary 4.2, we leverage these results and pre-existing Rademacher complexity bounds to conclude that the optimizer of the weakly-regularized logistic loss will have width-free generalization bound scaling with the inverse of the max margin and network depth. Finally, we note that the maximum possible margin can only increase with the width of the network, which suggests that increasing width can improve generalization of the solution (see Theorem 4.3).

We work with a family $\mathcal{F}$ of prediction functions $f(\cdot; \Theta) : \mathbb{R}^d \to \mathbb{R}$ that are $a$-positive-homogeneous in their parameters for some $a > 0$: $f(x; c\Theta) = c^a f(x; \Theta), \forall c > 0$. We additionally require that $f$ is continuous when viewed as a function in $\Theta$. For some general norm $\|\cdot\|$ and $\lambda > 0$, we study the $\lambda$-regularized logistic loss $L_\lambda$, defined as

$$L_\lambda(\Theta) \triangleq \frac{1}{n} \sum_{i=1}^n \log(1 + \exp(-y_i f(x_i; \Theta))) + \lambda \|\Theta\|^r \qquad (4.1)$$

for fixed $r > 0$. Let $\Theta_\lambda \in \arg\min L_\lambda(\Theta)$.[4] Define the normalized margin $\gamma_\lambda$ and max-margin $\gamma^\star$ by $\gamma_\lambda \triangleq \min_i y_i f(x_i; \bar{\Theta}_\lambda)$ and $\gamma^\star \triangleq \max_{\|\Theta\| \le 1} \min_i y_i f(x_i; \Theta)$. Let $\Theta^\star$ achieve this maximum.

We show that with sufficiently small regularization level $\lambda$, the normalized margin $\gamma_\lambda$ approaches the maximum margin $\gamma^\star$. Our theorem and proof are inspired by the result of Rosset et al. [58, 59], who analyze the special case when $f$ is a linear function. In contrast, our result can be applied to non-linear $f$ as long as $f$ is homogeneous.

**Theorem 4.1.** *Assume the training data is separable by a network $f(\cdot; \Theta^\star) \in \mathcal{F}$ with an optimal normalized margin $\gamma^\star > 0$. Then, the normalized margin of the global optimum of the weakly-regularized objective (equation 4.1) converges to $\gamma^\star$ as the regularization goes to zero. Mathematically,*

$$\gamma_\lambda \to \gamma^\star \text{ as } \lambda \to 0$$

An intuitive explanation for our result is as follows: because of the homogeneity, the loss $L(\Theta_\lambda)$ roughly satisfies the following (for small $\lambda$, and ignoring parameters such as $n$):

$$L_\lambda(\Theta_\lambda) \approx \exp(-\|\Theta_\lambda\|^a \gamma_\lambda) + \lambda \|\Theta_\lambda\|^r$$

Thus, the loss selects parameters with larger margin, while the regularization favors smaller norms. The full proof of the theorem is deferred to Section C.

Though the result in this section is stated for binary classification, it extends to the multi-class setting with cross-entropy loss. We provide formal definitions and results in Section C. In Theorem C.2, we also show that an approximate minimizer of $L_\lambda$ can obtain margin that approximates $\gamma^\star$.

Although we consider an *explicit* regularizer, our result is related to recent works on algorithmic regularization of gradient descent for the *unregularized* objective. Recent works show that gradient descent finds the minimum norm or max-margin solution for problems including logistic regression, linearized neural networks, and matrix factorization [68, 28, 38, 27, 32]. Many of these proofs require a delicate analysis of the algorithm's dynamics, and some are not fully rigorous due to assumptions on the iterates. To the best of our knowledge, it is an open question to prove analogous results for even two-layer relu networks. In contrast, by adding the explicit $\ell_2$ regularizer to our objective, we can prove broader results that apply to multi-layer relu networks. In the following section we leverage our result and existing generalization bounds [25] to help justify how over-parameterization can improve generalization.

### 4.1 Generalization of the Max-Margin Neural Net

We consider depth-$q$ networks with 1-Lipschitz, 1-positive-homogeneous activation $\phi$ for $q \geq 2$. Note that the network function is $q$-positive-homogeneous. Suppose that the collection of parameters $\Theta$ is given by matrices $W_1, \ldots, W_q$. For simplicity we work in the binary class setting, so the $q$-layer network computes a real-valued score

$$f^{\text{NN}}(x; \Theta) \triangleq W_q \phi(W_{q-1} \phi(\cdots \phi(W_1 x) \cdots)) \tag{4.2}$$

where we overload notation to let $\phi(\cdot)$ denote the element-wise application of the activation $\phi$. Let $m_i$ denote the size of the $i$-th hidden layer, so $W_1 \in \mathbb{R}^{m_1 \times d}, W_2 \in \mathbb{R}^{m_2 \times m_1}, \cdots, W_q \in \mathbb{R}^{1 \times m_{q-1}}$. We will let $\mathcal{M} \triangleq (m_1, \ldots, m_{q-1})$ denote the sequence of hidden layer sizes. We will focus on $\ell_2$-regularized logistic loss (see equation 4.1, using $\|\cdot\|_F$ and $r = 2$) and denote it by $L_{\lambda,\mathcal{M}}$.

Following notation established in this section, we denote the optimizer of $L_{\lambda,\mathcal{M}}$ by $\Theta_{\lambda,\mathcal{M}}$, the normalized margin of $\Theta_{\lambda,\mathcal{M}}$ by $\gamma_{\lambda,\mathcal{M}}$, the max-margin solution by $\Theta^{\star,\mathcal{M}}$, and the max-margin by $\gamma^{\star,\mathcal{M}}$, assumed to be positive. Our notation emphasizes the architecture of the network.

We can define the population 0-1 loss of the network parameterized by $\Theta$ by $L(\Theta) \triangleq \Pr_{(x,y) \sim p_{\text{data}}}[y f^{\text{NN}}(x; \Theta) \leq 0]$. We let $\mathcal{X}$ denote the data domain and $C \triangleq \sup_{x \in \mathcal{X}} \|x\|_2$ denote the largest possible norm of a single datapoint.

By combining the neural net complexity bounds of Golowich et al. [25] with our Theorem 4.1, we can conclude that optimizing weakly-regularized logistic loss gives generalization bounds that depend on the maximum possible network margin for the given architecture.

**Corollary 4.2.** *Suppose $\phi$ is 1-Lipschitz and 1-positive-homogeneous. With probability at least $1 - \delta$ over the draw of $(x_1, y_1), \ldots, (x_n, y_n)$ i.i.d. from $p_{\text{data}}$, we can bound the test error of the optimizer of the regularized loss by*

$$\limsup_{\lambda \to 0} L(\Theta_{\lambda,\mathcal{M}}) \lesssim \frac{C}{\gamma^{\star,\mathcal{M}} q^{\frac{q-1}{2}} \sqrt{n}} + \epsilon(\gamma^{\star,\mathcal{M}}) \tag{4.3}$$

*where $\epsilon(\gamma) \triangleq \sqrt{\frac{\log \log_2 \frac{4C}{\gamma}}{n}} + \sqrt{\frac{\log(1/\delta)}{n}}$. Note that $\epsilon(\gamma^{\star,\mathcal{M}})$ is primarily a smaller order term, so the bound mainly scales with $\frac{C}{\gamma^{\star,\mathcal{M}} q^{(q-1)/2} \sqrt{n}}$.* [5]

Finally, we observe that the maximum normalized margin is non-decreasing with the size of the architecture. Formally, for two depth-$q$ architectures $\mathcal{M} = (m_1, \ldots, m_{q-1})$ and $\mathcal{M}' = (m'_1, \ldots, m'_{q-1})$, we say $\mathcal{M} \leq \mathcal{M}'$ if $m_i \leq m'_i \ \forall i = 1, \ldots q - 1$. Theorem 4.3 states if $\mathcal{M} \leq \mathcal{M}'$, the max-margin over networks with architecture $\mathcal{M}'$ is at least the max-margin over networks with architecture $\mathcal{M}$.

**Theorem 4.3.** *Recall that $\gamma^{\star,\mathcal{M}}$ denotes the maximum normalized margin of a network with architecture $\mathcal{M}$. If $\mathcal{M} \leq \mathcal{M}'$, we have*

$$\gamma^{\star,\mathcal{M}} \leq \gamma^{\star,\mathcal{M}'}$$

*As a important consequence, the generalization error bound of Corollary 4.2 for $\mathcal{M}'$ is at least as good as that for $\mathcal{M}$.*

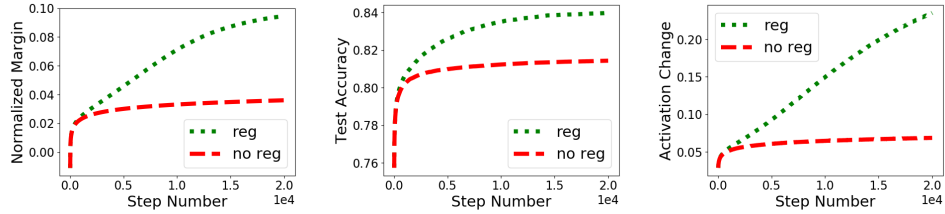

Figure 2: Comparing regularization and no regularization starting from the same initialization. **Left:** Normalized margin. **Center:** Test accuracy. **Right:** Percentage of activation patterns changed.

This theorem is simple to prove and follows because we can directly implement any network of architecture $\mathcal{M}$ using one of architecture $\mathcal{M}'$, if $\mathcal{M} \leq \mathcal{M}'$. This highlights one of the benefits of over-parametrization: the margin does not decrease with a larger network size, and therefore Corollary 4.2 gives a better generalization bound. In Section F, we provide empirical evidence that the test error decreases with larger network size while the margin is non-decreasing.

The phenomenon in Theorem 4.3 contrasts with standard $\ell_2$-normalized linear prediction. In this setting, adding more features increases the norm of the data, and therefore the generalization error bounds could also increase. On the other hand, Theorem 4.3 shows that adding more neurons (which can be viewed as learned features) can only improve the generalization of the max-margin solution.

## 5 Simulations

We empirically validate our theory with several simulations. First, we train a two-layer net on synthetic data with and without explicit regularization starting from the same initialization in order to demonstrate the effect of an explicit regularizer on generalization. We confirm that the regularized network does indeed generalize better and moves further from its initialization. For this experiment, we use a large initialization scale, so every weight $\sim \mathcal{N}(0, 1)$. We average this experiment over 20 trials and plot the test accuracy, normalized margin, and percentage change in activation patterns in Figure 2. We compute the percentage of activation patterns changed over every possible pair of hidden unit and training example. Since a low percentage of activations change when $\lambda = 0$, the unregularized neural net learns in the kernel regime. Our simulations demonstrate that an explicit regularizer improves generalization error as well as the margin, as predicted by our theory.

The data comes from a ground truth network with 10 hidden networks, input dimension 20, and a ground truth unnormalized margin of at least 0.01. We use a training set of size 200 and train for 20000 steps with learning rate 0.1, once using regularizer $\lambda = 5 \times 10^{-4}$ and once using regularization $\lambda = 0$. We note that the training error hits 0 extremely quickly (within 50 training iterations). The initial normalized margin is negative because the training error has not yet hit zero.

We also compare the generalization of a regularized neural net and kernel method as the sample size increases. Furthermore, we demonstrate that for two-layer nets, the test error decreases and margin increases as the width of the hidden layer grows, as predicted by our theory. We provide figures and full details in Section F.

## 6 Conclusion

We have shown theoretically and empirically that explicitly $\ell_2$ regularized neural nets can generalize better than the corresponding kernel method. We also argue that maximizing margin is one of the inductive biases of relu networks obtained from optimizing weakly-regularized cross-entropy loss. To complement these generalization results, we study optimization and prove that it is possible to find a global minimizer of the regularized loss in polynomial time when the network width is infinite. A natural direction for future work is to apply our theory to optimize the margin of finite-sized neural networks.

## Acknowledgments

CW acknowledges the support of a NSF Graduate Research Fellowship. JDL acknowledges support of the ARO under MURI Award W911NF-11-1-0303. This is part of the collaboration between US DOD, UK MOD and UK Engineering and Physical Research Council (EPSRC) under the Multidisciplinary University Research Initiative. We also thank Nati Srebro and Suriya Gunasekar for helpful discussions in various stages of this work.

## Footnotes

[1]The truncation is required to prove generalization of the regularized neural net using standard tools.

[2]There could be some variations of the NTK space depending on the scales of the initialization of the two layers, but our Theorem 2.1 shows that these variations also suffer from a worse sample complexity.

[3]The relu activation is non-differentiable at 0 and hence the gradient flow is not well-defined. Chizat and Bach [15] acknowledge this same difficulty with relu.

[4]We formally show that $L_\lambda$ has a minimizer in Claim C.3 of Section C.

[5]Although the $\frac{1}{q^{(q-1)/2}}$ factor of equation D.1 decreases with depth $q$, the margin $\gamma$ will also tend to decrease as the constraint $\|\bar{\Theta}\|_F \leq 1$ becomes more stringent.

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
