[Supplementary Material]

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

[6]The factor of $\frac{1}{2}$ is due the the relation that every unit-norm parameter $\Theta$ corresponds to an $\mu$ in the lifted space with $\|\mu\| = 2$.

[7]Although the $\frac{1}{K^{(K-1)/2}}$ factor of equation D.1 decreases with depth $K$, the margin $\gamma$ will also tend to decrease as the constraint $\|\bar{\Theta}\|_F \le 1$ becomes more stringent.

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

# A   Additional Notation

In this section we collect additional notations that will be useful for our proofs.

Let $\mathbb{S}^{d-1} \triangleq \{\bar{u} \in \mathbb{R}^d : \|\bar{u}\|_2 = 1\}$ be the unit sphere in $d$ dimensions. Let $\mathcal{L}_k^2(\mathbb{S}^{d-1})$ be the space of functions on $\mathbb{S}^{d-1} \to \mathbb{R}^k$ for which the squared $\ell_2$ norm of the function value is Lebesgue integrable. For $\varphi_1, \varphi_2 \in \mathcal{L}_k^2(\mathbb{S}^{d-1})$, we can define $\langle \varphi_1, \varphi_2 \rangle \triangleq \int_{\mathbb{S}^{d-1}} \varphi_1(\bar{u})^\top \varphi_2(\bar{u}) d\bar{u} < \infty$.

For general $p$, will also define $\mathcal{L}_1^p(\mathbb{S}^{d-1})$ be the space of functions on $\mathbb{S}^{d-1}$ for which the $p$-th power of the absolute value is Lebesgue integrable. For $\varphi \in \mathcal{L}_1^p(\mathbb{S}^{d-1})$, we overload notation and write $\|\varphi\|_p \triangleq \left( \int_{\mathbb{S}^{d-1}} |\varphi(\bar{u})|^p d\bar{u} \right)^{1/p}$. Additionally, for $\varphi_1 \in \mathcal{L}_1^1(\mathbb{S}^{d-1})$ and $\varphi_2 \in \mathcal{L}_1^\infty(\mathbb{S}^{d-1})$, we can define $\langle \varphi_1, \varphi_2 \rangle \triangleq \int_{\mathbb{S}^{d-1}} \varphi_1(\bar{u}) \varphi_2(\bar{u}) d\bar{u} < \infty$.

# B   Missing Material from Section 2

## B.1   Lower Bound on NTK Kernel Generalization

In this section we will lower bound the test error of the kernel prediction function for our distribution $\mathcal{D}$ in the setting of Theorem 2.1. We will first introduce some additional notation to facilitate the proofs in this section. Let $\mathcal{D}_x$ be the marginal distribution of $\mathcal{D}$ over datapoints $x$. We use $z_i$ to refer to the last $d-2$ coordinates of $x_i$. For a given vector $x$, $x_{-2}$ will index the last $d-2$ coordinates of a vector $x$ and for $z \in \mathbb{R}^{d-2}$, use $(a, b, z)$ to denote the vector in $\mathbb{R}^d$ with first two coordinates $a, b$, and last $d-2$ coordinates $z$. For a vector $x \in \mathbb{R}^d$, let $x^{\otimes 2} \in \mathbb{R}^{d^2}$ denote the vector with $(i-1)d + j$-th entry $e_i^\top x e_j^\top x$.

Furthermore, we define the following lifting functions $\varphi_{\text{grad}}, \varphi_{\text{relu}}$ mapping data $x \in \mathbb{R}^d$ to an infinite feature vector:

$$\varphi_{\text{grad}}(x) \in \mathcal{L}_d^2(\mathbb{S}^{d-1}) \text{ satisfies } \varphi_{\text{grad}}(x)[\bar{u}] = \mathbb{1}(x^\top \bar{u} \geq 0) x$$
$$\varphi_{\text{relu}}(x) \in \mathcal{L}_1^\infty(\mathbb{S}^{d-1}) \text{ satisfies } \varphi_{\text{relu}}(x)[\bar{u}] = [x^\top \bar{u}]_+$$

Note that the kernel $K(x', x)$ can be written as a sum of positive scalings of $\langle \varphi_{\text{grad}}(x), \varphi_{\text{grad}}(x') \rangle$ and $\langle \varphi_{\text{relu}}(x), \varphi_{\text{relu}}(x') \rangle$. We now define the following functions $K_1, K_2 : \mathbb{R}^d \times \mathbb{R}^d \mapsto \mathbb{R}$:

$$K_1(x', x) = x^\top x' \left( 1 - \pi^{-1} \arccos \left( \frac{x^\top x'}{\|x\|_2 \|x'\|_2} \right) \right)$$

$$K_2(x', x) = \frac{\|x\|_2 \|x'\|_2}{\pi} \sqrt{1 - \left( \frac{x^\top x'}{\|x\|_2 \|x'\|_2} \right)^2}$$

We have

$$\langle \varphi_{\text{grad}}(x), \varphi_{\text{grad}}(x') \rangle = K_1(x', x)$$
$$\langle \varphi_{\text{relu}}(x), \varphi_{\text{relu}}(x') \rangle = c_{\text{relu}}(K_1(x', x) + K_2(x', x))$$

for some $c_{\text{relu}} > 0$. The second equation follows from Lemma A.1 of [20]. To see the first one, we note that the indicator $\mathbb{1}(x'^\top \bar{u} \geq 0) \mathbb{1}(x^\top \bar{u} \geq 0)$ is only 1 in a arc of degree $\pi - \arccos(x^\top x'/\|x\|_2 \|x'\|_2)$ between $x$ and $x'$. As all directions are equally likely, the expectation $\mathbb{E}_{\bar{u}}[\mathbb{1}(x'^\top \bar{u} \geq 0) \mathbb{1}(x^\top \bar{u} \geq 0)] = 1 - \pi^{-1} \arccos \left( \frac{x^\top x'}{\|x\|_2 \|x'\|_2} \right)$.

Then as the kernel $K(x', x)$ is the sum of positive scalings of $\langle \varphi_{\text{grad}}(x), \varphi_{\text{grad}}(x') \rangle$ and $\langle \varphi_{\text{relu}}(x), \varphi_{\text{relu}}(x') \rangle$, we can express

$$K(x', x) = \tau_1 K_1(x', x) + \tau_2(K_1(x', x) + K_2(x', x)) \tag{B.1}$$

for $\tau_1, \tau_2 > 0$. This decomposition will be useful in our analysis of the lower bound. The following theorem restates our lower bound on the test error of any $\ell_2$-regularized kernel method.

**Theorem B.1.** *For the distribution $\mathcal{D}$ defined in Section 2, if $n \lesssim d^2$, with probability $1 - \exp(-\Omega(\sqrt{n}))$ over $(x_1, y_1), \ldots, (x_n, y_n)$ drawn i.i.d. from $\mathcal{D}$, for all choices of $\beta$, in test time the kernel prediction function $f^{\text{kernel}}(\cdot; \beta)$ will predict the sign of $y$ wrong $\Omega(1)$ fraction of the time:*

$$\Pr_{(x,y) \sim \mathcal{D}} [f^{\text{kernel}}(x; \beta) y \leq 0] = \Omega(1)$$

As it will be clear from context, we drop the $^{\text{kernel}}$ superscript. The first step of our proof will be demonstrating that the first two coordinates do not affect the value of the prediction function $f(x; \beta)$ by very much. This is where we formalize the importance of having the sign of the positive label be unaffected by the sign of the first coordinate, and likewise for the second coordinate and negative labels. We utilize the sign symmetry to induce further cancellations in the prediction function output. Formally, we will first define the functions $\tilde{K}_1, \tilde{K}_2 : \mathbb{R}^{d-2} \times \mathbb{R}^{d-2} \mapsto \mathbb{R}$ with

$$\tilde{K}_1(z', z) = K_1((0, 1, z'), (1, 0, z))$$
$$\tilde{K}_2(z', z) = K_2((0, 1, z'), (1, 0, z))$$

Next, we will define the function $\tilde{f} : \mathbb{R}^{d-2} \mapsto \mathbb{R}$ with

$$\tilde{f}(z; \beta) = \tau_1 \sum_{i=1}^{n} \beta_i \tilde{K}_1(z_i, z) + \tau_2 \sum_{i=1}^{n} \beta_i (\tilde{K}_1(z_i, z) + \tilde{K}_2(z_i, z))$$

The following lemma states that $2\tilde{f}(z; \beta)$ will approximate both $f((1, 0, z); \beta) + f((-1, 0, z); \beta)$ and $f((0, 1, z); \beta) + f((0, -1, z); \beta)$. This allows us to immediately lower bound the test error of $f$ by the probability that $\tilde{f}(z; \beta)$ is sufficiently large.

**Lemma B.2.** *Define the functions*

$$f^+(z; \beta) \triangleq f((1, 0, z); \beta) + f((-1, 0, z); \beta)$$
$$f^-(z; \beta) \triangleq f((0, 1, z); \beta) + f((0, -1, z); \beta)$$

*Then with probability $1 - \exp(-\Omega(d))$, there is some universal constant $c$ such that*

$$|f^+(z; \beta) - 2\tilde{f}(z; \beta)| \leq \frac{c(\tau_1 + \tau_2)}{d} \sum_{i=1}^{n} |\beta_i|$$

$$|f^-(z; \beta) - 2\tilde{f}(z; \beta)| \leq \frac{c(\tau_1 + \tau_2)}{d} \sum_{i=1}^{n} |\beta_i|$$

(B.2)

*As a result, for all choices of $\beta_1, \ldots, \beta_n$, we can lower bound the test error of the kernel prediction function by*

$$\Pr_{(x,y) \sim \mathcal{D}}[f(x; \beta) y \leq 0] \geq \frac{1}{4} \Pr_{z \sim \{-1, +1\}^{d-2}} \left( |\tilde{f}(z; \beta)| \geq \frac{3c(\tau_1 + \tau_2)}{2d} \sum_{i=1}^{n} |\beta_i| \right) - \exp(-\Omega(d))$$

Now we argue that $|\tilde{f}(z; \beta)|$ will be large with constant probability over $z$, leading to constant test error of $f$. Formally we first show that with constant probability over the choice of $z \sim \{-1, +1\}^{d-2}$, we have $|\tilde{f}(z; \beta)| \geq \frac{3c(\tau_1 + \tau_2)}{2d} \sum_{i=1}^{n} |\beta_i|$.

**Lemma B.3.** *For sufficiently small $n \lesssim d^2$, with probability $1 - \exp(-\Omega(\sqrt{n}))$ over the random draws of $z_1, \ldots, z_n$, the following holds: for all $\beta_1, \ldots, \beta_n$, we will have*

$$\Pr_{z \sim \{-1, +1\}^{d-2}} \left( |\tilde{f}(z; \beta)| \geq \frac{3c(\tau_1 + \tau_2)}{2d} \sum_{i=1}^{n} |\beta_i| \right) \geq \Omega(1)$$

*where $c$ is the constant defined in Lemma B.2.*

This will allow us to complete the proof of Theorem B.1.

*Proof of Theorem B.1.* By plugging Lemma B.3 into the statement of Lemma B.2, we can conclude that for sufficiently small $n \lesssim d^2$, with probability $1 - \exp(-\Omega(\sqrt{n}))$ over the random draws of $z_1, \ldots, z_n$, we have

$$\Pr_{(x,y) \sim \mathcal{D}}[f(x; \beta) y \leq 0] \geq \Omega(1)$$

for all choices of $\beta$. This gives precisely Theorem B.1. $\square$

It now suffices to prove Lemmas B.2 and B.3.

To prove Lemma B.2, we will rely on the following two lemmas relating $K_1, K_2$ with $\tilde{K}_1, \tilde{K}_2$, stated and proved below:

**Lemma B.4.** *Let $z \in \{-1, +1\}^{d-2}$ be a uniform random point from the $d-2$-dimensional hypercube and $x \in supp(\mathcal{D}_x)$ be given. With probability $1 - \exp(-\Omega(d))$ over the choice of $z$, we have*

$$|K_1(x, (1, 0, z)) + K_1(x, (-1, 0, z)) - 2\tilde{K}_1(x_{-2}, z)| \lesssim \frac{1}{d}$$

$$|K_1(x, (0, 1, z)) + K_1(x, (0, -1, z)) - 2\tilde{K}_1(x_{-2}, z)| \lesssim \frac{1}{d}$$

**Lemma B.5.** *In the same setting as Lemma B.4, with probability $1 - \exp(-\Omega(d))$ over the choice of $z$, we have*

$$|K_2(x, (1, 0, z)) + K_2(x, (-1, 0, z)) - 2\tilde{K}_2(x_{-2}, z)| \lesssim \frac{1}{d}$$

$$|K_2(x, (0, 1, z)) + K_2(x, (0, -1, z)) - 2\tilde{K}_2(x_{-2}, z)| \lesssim \frac{1}{d}$$

*Proof of Lemma B.4.* As it will be clear in the context of this proof, we use $x_1$ to denote the first coordinate of $x$ and $x_2$ to denote the second coordinate of $x$. We prove the first inequality, as the proof for the second is identical. First, note that if $x_1 = 0, |x_2| = 1$, then we have $K_1(x, (1, 0, z)) + K_1(x, (-1, 0, z)) = 2K_1((0, 1, x_{-2}), (1, 0, z))$ so the inequality holds trivially. Thus, we work in the case that $|x_1| = 1, x_2 = 0$.

Note that $\|(1, 0, z)\|_2 = \|(-1, 0, z)\|_2 = \|x\|_2 = \sqrt{d-1}$. We have:

$$K_1(x, (1, 0, z)) + K_1(x, (-1, 0, z)) \tag{B.3}$$

$$= \left(1 - \pi^{-1} \arccos\left(\frac{1 + x_{-2}^\top z}{d-1}\right)\right)(1 + x_{-2}^\top z)$$

$$+ \left(1 - \pi^{-1} \arccos\left(\frac{-1 + x_{-2}^\top z}{d-1}\right)\right)(-1 + x_{-2}^\top z)$$

$$= \pi^{-1}\left(\arccos\left(\frac{-1 + x_{-2}^\top z}{d-1}\right) - \arccos\left(\frac{1 + x_{-2}^\top z}{d-1}\right)\right) \tag{B.4}$$

$$+ x_{-2}^\top z \left(2 - \pi^{-1}\arccos\left(\frac{-1 + x_{-2}^\top z}{d-1}\right) - \pi^{-1}\arccos\left(\frac{1 + x_{-2}^\top z}{d-1}\right)\right) \tag{B.5}$$

Now we perform a Taylor expansion of $\arccos$ around $\nu \triangleq x_{-2}^\top z / (d-1)$ to get

$$\arccos(\nu + \epsilon) = \arccos(\nu) + \arccos'(\nu)\epsilon + O(\epsilon^2)$$

for any $|\nu|, |\nu + \epsilon| \le 3/4$. Note that this happens with probability $1 - \exp(-\Omega(d))$ by Hoeffding's inequality. Furthermore, for $|\nu| \le 3/4$, $\arccos'(\nu) = O(1)$, so we get that equation B.4 can be bounded by $O(\frac{1}{d})$. Next, we claim the following:

$$\left|\arccos\left(\frac{-1 + x_{-2}^\top z}{d-1}\right) + \arccos\left(\frac{1 + x_{-2}^\top z}{d-1}\right) - 2\arccos\left(\frac{x_{-2}^\top z}{d-1}\right)\right| = O\left(\frac{1}{d^2}\right)$$

This follows simply from Taylor expansion around $\nu$ setting $\epsilon$ to $\pm\frac{1}{d-1}$. Substituting this into equation B.5 and using our bound on equation B.4, we get

$$\left|K_1(x, (1, 0, z)) + K_1(x, (-1, 0, z)) - 2x_{-2}^\top z\left(1 - \pi^{-1}\arccos\left(\frac{x_{-2}^\top z}{d-1}\right)\right)\right| \le O\left(\frac{1}{d}\right)$$

Now we use the fact that $x_{-2}^\top z\left(1 - \pi^{-1}\arccos\left(\frac{x_{-2}^\top z}{d-1}\right)\right) = K_1((0, 1, x_{-2}), (1, 0, z))$ to complete the proof. $\qquad\square$

*Proof of Lemma B.5.* As before, it suffices to prove the first inequality in the case that $|x_1| = 1$, $x_2 = 0$. We can compute

$$(K_2(x, (1, 0, z)) + K_2(x, (-1, 0, z)) =$$

$$\frac{1}{\pi}\left((d-1)\sqrt{1 - \left(\frac{1 + x_{-2}^\top z}{d-1}\right)^2} + (d-1)\sqrt{1 - \left(\frac{-1 + x_{-2}^\top z}{d-1}\right)^2}\right) \tag{B.6}$$

Now we again perform a Taylor expansion, this time of $g(v) = \sqrt{1 - v^2}$ around $\nu \triangleq \frac{x_{-2}^\top z}{d-1}$. We get

$$g(\nu + \epsilon) = g(\nu) + g'(\nu)\epsilon + O(\epsilon^2)$$

for any $|\nu|, |\nu + \epsilon| \leq 3/4$. Note that $|\nu|, |\nu + \epsilon| \leq 3/4$ with probability $1 - \exp(-\Omega(d))$ via straightforward concentration. It follows that

$$\left|\sqrt{1 - \left(\frac{1 + x_{-2}^\top z}{d-1}\right)^2} + \sqrt{1 - \left(\frac{-1 + x_{-2}^\top z}{d-1}\right)^2} - 2\sqrt{1 - \left(\frac{x_{-2}^\top z}{d-1}\right)^2}\right| \lesssim \frac{1}{d^2}$$

Now plugging this into equation B.6 and using the fact that $\frac{1}{\pi}(d-1)\sqrt{1 - \left(\frac{x_{-2}^\top z}{d-1}\right)^2} = K_2((0, 1, x_{-2}), (1, 0, z))$ gives the desired result. $\qquad\square$

Now we can complete the proof of Lemma B.2.

*Proof of Lemma B.2.* We note that

$$|f^+(z; \beta) - 2\tilde{f}(z; \beta)| = \left|(\tau_1 + \tau_2)\sum_{i=1}^n \beta_i[K_1((1, 0, z), x_i) + K_1((-1, 0, z), x_i) - 2\tilde{K}_1(z_i, z)]\right.$$

$$\left. + \tau_2\sum_{i=1}^n \beta_i[K_2((1, 0, z), x_i) + K_2((-1, 0, z), x_i) - 2\tilde{K}_2(z_i, z)]\right| \tag{B.7}$$

Now with applying Lemmas B.4 and B.5 with a union bound over all $i$, we get with probability $1 - \exp(-\Omega(d))$ over the choice of $z$ uniform from $\{-1, +1\}^{d-2}$, for all $i$

$$|K_1((1, 0, z), x_i) + K_1((-1, 0, z), x_i) - 2\tilde{K}_1(z_i, z)| \lesssim \frac{1}{d}$$

$$|K_2((1, 0, z), x_i) + K_2((-1, 0, z), x_i) - 2\tilde{K}_2(z_i, z)| \lesssim \frac{1}{d}$$

Now plugging into equation B.7 and applying triangle inequality gives us

$$|f^+(z; \beta) - 2\tilde{f}(z; \beta)| \leq \frac{c(\tau_1 + \tau_2)}{d}\sum_{i=1}^n |\beta_i| \tag{B.8}$$

with probablity $1 - \exp(-\Omega(d))$ over $z$ for some universal constant $c$. An identical argument also gives us

$$|f^-(z; \beta) - 2\tilde{f}(z; \beta)| \leq \frac{c(\tau_1 + \tau_2)}{d}\sum_{i=1}^n |\beta_i| \tag{B.9}$$

Finally, to lower bound the quantity $\Pr_{(x,y)\sim\mathcal{D}}[f(x; \beta)y \leq 0]$, we note that if

$$|\tilde{f}(z; \beta)| \geq \frac{3c(\tau_1 + \tau_2)}{2d}\sum_{i=1}^n |\beta_i|$$

and equation B.2 hold, then $f^+(z; \beta)$ and $f^-(z; \beta)$ will have the same sign. However, this in turn means that one of the following must hold:

$$f((1, 0, z); \beta) < 0$$
$$f((-1, 0, z); \beta) < 0$$
$$f((0, 1, z); \beta) > 0$$
$$f((0, -1, z); \beta) > 0$$

which implies an incorrect predicted sign. As $(1, 0, z)$, $(-1, 0, z)$, $(0, 1, z)$, $(0, -1, z)$ are all equally likely under distribution $\mathcal{D}_x$, the probability of drawing one of these examples under $\mathcal{D}_x$ is at least

$$\frac{1}{4} \Pr_{z \sim \{-1, +1\}^{d-2}} \left( |\tilde{f}(z; \beta)| \geq \frac{3c(\tau_1 + \tau_2)}{2d} \sum_{i=1}^{n} |\beta_i| \right) - \exp(-\Omega(d))$$

This gives the desired lower bound on $\Pr_{(x,y) \sim \mathcal{D}}[f(x; \beta)y \leq 0]$. $\qquad \square$

Now we will prove Lemma B.3. We will first construct a polynomial approximation $\hat{f}(z; \beta)$ of $\tilde{f}(z; \beta)$, and then lower bound the expectation $\mathbb{E}_z[\hat{f}(z; \beta)^2]$. We use the following two lemmas:

**Lemma B.6.** *Define the polynomial $g : \mathbb{R} \mapsto \mathbb{R}$ as follows:*

$$g(x) \triangleq \tau_1(d-1) \left( \frac{1}{2}x + \frac{1}{\pi}x^2 + \frac{1}{6\pi}x^4 \right) + \tau_2(d-1) \left( \frac{1}{\pi} + \frac{1}{2}x + \frac{1}{2\pi}x^2 + \frac{1}{24\pi}x^4 \right)$$

*Then for $z \in \{-1, +1\}^{d-2}$ distributed uniformly over the hypercube and some given $z' \in \{-1, +1\}^{d-2}$,*

$$\Pr_z \left[ \left| g\left(\frac{z^\top z'}{d-1}\right) - (\tau_1 + \tau_2)\tilde{K}_1(z, z') - \tau_2\tilde{K}_2(z, z') \right| \leq c_1(\tau_1 + \tau_2)\frac{\log^{2.5}}{d^{1.5}} \right] \geq 1 - d^{-10}$$

*for some universal constant $c_1$.*

**Lemma B.7.** *Let $g : \mathbb{R} \mapsto \mathbb{R}$ be any degree-$k$ polynomial with nonnegative coefficients, i.e. $g(x) = \sum_{j=1}^{k} a_j x^j$ with $a_j \geq 0$ for all $j$. For $n \lesssim d^2$, with probability $1 - \exp(-\Omega(\sqrt{n}))$ over the random draws of $z_1, \dots, z_n$ i.i.d. uniform from $\{-1, +1\}^d$, the following holds: for all $\beta_1, \dots, \beta_n$, we will have*

$$\mathbb{E}_z \left[ \left( \sum_{i=1}^{n} \beta_i g(z^\top z_i) \right)^2 \right] \gtrsim a_2^2 d^2 \sum_{i=1}^{n} \beta_i^2$$

*where $z \in \{-1, +1\}^d$ is a uniform vector from the hypercube.*

Now we provide the proof of Lemma B.3.

*Proof of Lemma B.3.* For the degree-4 polynomial $g$ defined in Lemma B.6, we define

$$\hat{f}(z; \beta) = \sum_{i=1}^{n} \beta_i g\left(\frac{z^\top z_i}{d-1}\right)$$

Note that with probability $1 - d^{-8}$ over the choice of $z$, $|\hat{f}(z; \beta) - \tilde{f}(z; \beta)| \lesssim \frac{\log^{2.5} d}{d^{1.5}}(\tau_1 + \tau_2) \sum_{i=1}^{n} |\beta_i|$.

With the purpose of applying Lemma B.7, we can first compute the coefficent of $x^2$ in $g(x/(d-1))$ to be $\frac{1}{\pi(d-1)}(\tau_1 + \tau_2/2)$. As $g$ has positive coefficients, we can thus apply Lemma B.7 to conclude that with high probability over $z_1, \dots, z_n$, the following event $\mathcal{E}$ holds: for all choices of $\beta_1, \dots, \beta_n$, $\mathbb{E}_z[\hat{f}(z; \beta)^2] \geq c_2(\tau_1 + \tau_2)^2 \sum_{i=1}^{n} \beta_i^2$ for some universal constant $c_2$. We now condition on the event that $\mathcal{E}$ holds.

Note that by Cauchy-Schartz, $\sum_{i=1}^{n} \beta_i^2 \geq \frac{1}{n}(\sum_{i=1}^{n} |\beta_i|)^2$. It follows that if $n \leq \frac{c_2}{4c^2}d^2$, we have

$$\mathbb{E}_z[\hat{f}(z; \beta)^2] \geq c_2(\tau_1 + \tau_2)^2 \sum_{i=1}^{n} \beta_i^2 \geq \frac{c_2(\tau_1 + \tau_2)^2}{n}(\sum_{i=1}^{n} |\beta_i|)^2 \geq \frac{4c^2(\tau_1 + \tau_2)^2}{d^2}(\sum_{i=1}^{n} |\beta_i|)^2$$

Now we can apply Bonami's Lemma (see Chapter 9 of O'Donnell [57]) along with the fact that $\hat{f}$ is a degree-4 polynomial in i.i.d. $\pm 1$ variables $z_1, \ldots, z_{d-2}$ to obtain

$$\mathbb{E}_z[\hat{f}(z;\beta)^4] \leq 9^4(\mathbb{E}_z[\hat{f}(z;\beta)^2])^2$$

Combining this with Proposition 9.4 of O'Donnell [57] lets us conclude that if $\mathcal{E}$ holds, with probability $\Omega(1)$ over the random draw of $z$,

$$|\hat{f}(z;\beta)| \geq \frac{3}{4}\sqrt{\mathbb{E}_z[\hat{f}(z;\beta)^2]} \geq \frac{3c(\tau_1 + \tau_2)}{2d} \sum_{i=1}^n |\beta_i|$$

Since $|\hat{f}(z;\beta) - \tilde{f}(z;\beta)| \lesssim \frac{(\tau_1 + \tau_2)\log^{2.5}(d)}{d^{1.5}} \sum_{i=1}^n |\beta_i|$ w.h.p over $z$, we can conclude that

$$|\tilde{f}(z;\beta)| \geq \frac{3c(\tau_1 + \tau_2)}{2d} \sum_{i=1}^n |\beta_i|$$

holds with probability $\Omega(1)$ over $z$. This gives the desired result. $\qquad\square$

*Proof of Lemma B.6.* Define functions $h_1, h_2 : (-1, 1) \mapsto \mathbb{R}$ with

$$h_1(x) = x(1 - \pi^{-1}\arccos x)$$
$$h_2(x) = \frac{1}{\pi}\sqrt{1 - x^2}$$

Recalling our definitions of $\tilde{K}_1, \tilde{K}_2$, it follows that $\tilde{K}_1(z, z') = (d-1)h_1\left(\frac{z^\top z'}{d-1}\right)$ and $\tilde{K}_2(z, z') = (d-1)h_2\left(\frac{z^\top z'}{d-1}\right)$. Letting $g_1, g_2$ denote the 4-th order Taylor expansions around 0 of $h_1, h_2$, respectively, it follows from straightforward calculation that

$$g_1(x) = \frac{1}{2}x + \frac{1}{\pi}x^2 + \frac{1}{6\pi}x^4$$
$$g_2(x) = \frac{1}{\pi} - \frac{1}{2\pi}x^2 - \frac{1}{8\pi}x^4$$

with $|h_1(x) - g_1(x)| \leq O(|x|^5)$ and $|h_2(x) - g_2(x)| \leq O(|x|^5)$ for $|x| \leq 3/4$. )Now we can observe that $g(x) = (\tau_1 + \tau_2)(d-1)g_1(x) + \tau_2(d-1)g_2(x)$. Thus,

$$g\left(\frac{z^\top z'}{d-1}\right) - (\tau_1 + \tau_2)\tilde{K}_1(z, z') - \tau_2\tilde{K}_2(z, z')$$

$$= (d-1)\left[(\tau_1 + \tau_2)\left(g_1\left(\frac{z^\top z'}{d-1}\right) - h_1\left(\frac{z^\top z'}{d-1}\right)\right) + \tau_2\left(g_2\left(\frac{z^\top z'}{d-1}\right) - h_2\left(\frac{z^\top z'}{d-1}\right)\right)\right]$$

As $|z^\top z'|/(d-1) \leq 3/4$ with probability $1 - \exp(-\Omega(d))$, the above is bounded in absolute value by $(d-1)(\tau_1 + \tau_2)O\left(\left(\frac{|z^\top z'|}{d-1}\right)^5\right)$. Finally, by Hoeffding's inequality $|z^\top z'| \leq c\sqrt{d\log d}$ with probability $1 - d^{-10}$ for some universal constant $c$. This gives the desired bound. $\qquad\square$

*Proof of Lemma B.7.* We first compute

$$\mathbb{E}_z\left[\left(\sum_{i=1}^n \beta_i g(z^\top z_i)\right)^2\right] = \mathbb{E}_z\left[\left(\sum_{i=1}^n \beta_i \sum_{j=1}^k a_j(z^\top z_i)^j\right)^2\right]$$

$$= \mathbb{E}_z\left[\left(\sum_{j=1}^k a_j \sum_{i=1}^n \beta_i(z^\top z_i)^j\right)^2\right]$$

$$= \sum_{j_1, j_2} a_{j_1}a_{j_2}\mathbb{E}_z\left[\left(\sum_{i=1}^n \beta_i(z^\top z_i)^{j_1}\right)\left(\sum_{i=1}^n \beta_i(z^\top z_i)^{j_2}\right)\right]$$

(expanding the square and using linearity of expectation)

Now note that all terms in the above sum are nonnegative by Lemma B.9 and the fact that $a_{j_1}, a_{j_2} \geq 0$. Thus, we can lower bound the above by the term corresponding to $j_1 = j_2 = 2$:

$$\mathbb{E}_z\left[\left(\sum_{i=1}^n \beta_i g(z^\top z_i)\right)^2\right] \geq a_2^2 \mathbb{E}_z\left[\left(\sum_{i=1}^n \beta_i (z^\top z_i)^2\right)\left(\sum_{i=1}^n \beta_i (z^\top z_i)^2\right)\right]$$

Now we can express

$$\mathbb{E}_z\left[\left(\sum_{i=1}^n \beta_i (z^\top z_i)^2\right)\left(\sum_{i=1}^n \beta_i (z^\top z_i)^2\right)\right] = \beta^\top M^{\otimes 2^\top} \mathbb{E}_z[z^{\otimes 2} z^{\otimes 2^\top}] M^{\otimes 2} \beta \qquad \text{(B.10)}$$

where $M \in \mathbb{R}^{d \times n}$ is the matrix with $z_i$ as its columns, and $M^{\otimes 2}$ has $z_i^{\otimes 2}$ as its columns.

We first compute $\mathbb{E}_z[z^{\otimes 2} z^{\otimes 2^\top}]$. Note that the entry in the $d(i_1 - 1) + j_1$-th row and $d(i_2 - 1) + j_2$-th column of $z^{\otimes 2} z^{\otimes 2^\top}$ is given by $(e_{i_1}^\top z)(e_{j_1}^\top z)(e_{i_2}^\top z)(e_{j_2}^\top z)$. Note that unless $i_1 = i_2$, $j_1 = j_2$ or $i_1 = j_1, i_2 = j_2$, this value has expectation 0. Thus, $\mathbb{E}_z[z^{\otimes 2} z^{\otimes 2^\top}]$ is a matrix with 1 on its diagonals and entries in the $(i-1)d + i$-th row and $(j-1)d + j$-th column, and 0 everywhere else. Letting $S$ denote the set of indices $\{(i-1)d + i : i \in [d]\}$ and $\vec{1}_S$ denote the vector in $\mathbb{R}^{d^2}$ with ones on $S$ and 0 everywhere else, we thus have

$$\mathbb{E}_z[z^{\otimes 2} z^{\otimes 2^\top}] = \vec{1}_S \vec{1}_S^\top + I_{[d^2]\setminus S \times [d^2]\setminus S}$$

Now letting $M_S^{\otimes 2}$ denote $M^{\otimes 2}$ with rows whose indices are not in $S$ zero'ed out, it follows that

$$M^{\otimes 2^\top} \mathbb{E}_z[z^{\otimes 2} z^{\otimes 2^\top}] M^{\otimes 2} = M_S^{\otimes 2^\top} \vec{1}_S \vec{1}_S^\top M_S^{\otimes 2} + M_{[d^2]\setminus S}^{\otimes 2^\top} I_{[d^2]\setminus S \times [d^2]\setminus S} M_{[d^2]\setminus S}^{\otimes 2}$$

$$\succeq M_{[d^2]\setminus S}^{\otimes 2^\top} M_{[d^2]\setminus S}^{\otimes 2} \qquad \text{(B.11)}$$

Therefore, it suffices to show $\sigma_{\min}(M_{[d^2]\setminus S}^{\otimes 2^\top} M_{[d^2]\setminus S}^{\otimes 2}) \gtrsim d^2$ with high probability. To do this, we can simply invoke Proposition 7.9 of Soltanolkotabi et al. [66] using $\eta_{\min} = \eta_{\max} = \sqrt{d^2 - d}$ and the fact that the columns of $M_{[d^2]\setminus S}^{\otimes 2}$ are $O(1)$-sub-exponential (Claim B.8 to get that if $n \leq cd^2$ for some universal constant $c$, then $\sigma_{\min}^2(M_{[d^2]\setminus S}^{\otimes 2}) \gtrsim d^2$ with probability $1 - \exp(O(\sqrt{n}))$.

Finally, combining this with equation B.11 and equation B.10 gives the desired result. $\qquad \square$

**Claim B.8.** *Say that a random vector $x \in \mathbb{R}^d$ is B-sub-exponential if the following holds:*

$$\sup_{y \in \mathbb{S}^{d-1}} \inf\{C > 0 : \mathbb{E}\exp(|x^\top y|/C) \leq 2\} \leq B$$

*Suppose that $z \sim \{-1, +1\}^d$ is a uniform vector on the hypercube. Then there is a universal constant $c$ such that $z^{\otimes 2} - \vec{1}_S$ is c-sub-exponential, where $S \triangleq \{(i-1)d + i : i \in [d]\}$ is the set of indices corresponding to squared entries of $z^{\otimes 2}$.*

*Proof.* Let $\tilde{z}^{\otimes 2}$ denote the $d^2 - d$ dimensional vector which removes coordinates in $S$ from $z^{\otimes 2}$. As $z^{\otimes 2}$ has value 1 with probability 1 on coordinates in $S$, it suffices to show that $\tilde{z}^{\otimes 2}$ is c-sub-exponential. We first note that for any $y \in \mathbb{R}^{d^2 - d}$, $y^\top \tilde{z}^{\otimes 2}$ can be written as $z^\top Y z$, where $Y$ is a $d \times d$ matrix with 0 on its diagonals and $ij$-th entry matching the corresponding entry of $y$.

Now we can apply Theorem 1.1 of Rudelson et al. [62], using the fact that $e_i^\top z$ have sub-Gaussian norm 2 to get

$$\Pr[|z^\top Y z| > t] \leq 2\exp(-c't^2/16\|y\|_2^2)$$

for some universal constant $c'$. Since this holds for all $y$, we can conclude the claim statement using Lemma 5.5 of Soltanolkotabi et al. [66]. $\qquad \square$

The following lemma is useful for proving the lower bound in Lemma B.7.

**Lemma B.9.** *Let $z_i \in \{-1, +1\}^d$ for $i \in [n]$, and let $z \in \{-1, +1\}^d$ be a vector sampled uniformly from the hypercube. Then for any integers $p, q \geq 0$,*

$$\mathbb{E}_z\left[\left(\sum_i \beta_i(z^\top z_i)^q\right)\left(\sum_i \beta_i(z^\top z_i)^p\right)\right] \geq 0$$

*Furthermore, equality holds if exactly one of $p$ or $q$ is odd.*

In order to prove Lemma B.9, we will require some tools and notation from boolean function analysis (see O'Donnell [57] for a more in-depth coverage). We first introduce the following notation: for $x \in \{-1, +1\}^d$ and $S \subseteq [d]$, we use $x^S$ to denote $\prod_{s \in S} x_s$. Then by Theorem 1.1 of [57], we can expand a function $f : \{-1, +1\}^d \mapsto \mathbb{R}$ with respect to the values $x^S$:

$$f(x) = \sum_{S \subseteq [d]} \hat{f}(S) x^S$$

where $\hat{f}(S)$ is called the Fourier coefficient of $f$ on $S$ and $\hat{f}(S) = \mathbb{E}_x[f(x)x^S]$ for $x$ uniform on $\{-1, +1\}^d$. For functions $f_1, f_2 : \{-1, +1\}^d \mapsto \mathbb{R}$, the following identity holds:

$$\mathbb{E}_x[f_1(x)f_2(x)] = \sum_{S \subseteq [d]} \hat{f}_1(S)\hat{f}_2(S) \tag{B.12}$$

*Proof of Lemma B.9.* For this proof we will use double indices on the $z_i$ vectors, so that $z_{i,j}$ will denote the $j$-th coordinate of $z_i$. We will only use the symbols $j$ to index the vectors $z, z_1, \ldots, z_n$. We define the functions $g(z) \triangleq \sum_i \beta_i(z^\top z_i)^q$ and $h(z) \triangleq \sum_i \beta_i(z^\top z_i)^p$, with Fourier coefficients $\hat{g}, \hat{h}$, respectively, and $g_i(z) = (z^\top z_i)^q$, $h_i(z) = (z^\top z_i)^p$ with Fourier coefficients $\hat{g}_i, \hat{h}_i$. We claim that for any $S \subseteq [d]$, $\hat{g}(S)\hat{h}(S) \geq 0$.

To see this, we will first compute $\hat{g}_i(S)$ as follows: $\hat{g}_i(S) = \mathbb{E}_z[(z^\top z_i)^q z^S]$. Now note that if we expand $(z^\top z_i)^q$ and compute this expectation, only terms of the form $z^S z_i^S z_{j_1}^{a_1} \cdots z_{j_k}^{a_k} z_{i,j_1}^{a_1} \cdots z_{i,j_k}^{a_k}$ with $a_1, \ldots, a_k$ even and $a_1 + \cdots + a_k = q - |S|$ are nonzero. Note that we have allowed $k$ to vary. Thus,

$$\mathbb{E}_z[(z^\top z_i)^q z^S] = \sum_{j_1, \ldots j_k, a_1, \ldots, a_k} (z^S)^2 z_i^S z_{j_1}^{a_1} \cdots z_{j_k}^{a_k} z_{i,j_1}^{a_1} \cdots z_{i,j_k}^{a_k}$$

$$= c_{q,|S|} z_i^S \tag{B.13}$$

for some positive integer $c_{q,|S|}$ depending only on $q, |S|$. We obtained equation B.13 via symmetry and the fact that $(z^S)^2 = 1$, $z_{j_1}^{a_1} \cdots z_{j_k}^{a_k} z_{i,j_1}^{a_1} \cdots z_{i,j_k}^{a_k} = 1$, as they are squares of values in $\{-1, +1\}$. Note that $c_{q,|S|} = 0$ for $|S| > q$. It follows that $\hat{g}(S) = c_{q,|S|} \sum_i \beta_i z_i^S$, and $\hat{h}(S) = c_{p,|S|} \sum_i \beta_i z_i^S$. Thus, $\hat{g}(S)\hat{h}(S) \geq 0 \forall S$, which means by equation B.12, we get

$$\mathbb{E}_z[g(z)h(z)] = \sum_S \hat{g}(S)\hat{h}(S) \geq 0$$

as desired.

Now to see that $\mathbb{E}_z[g(z)h(z)] = 0$ if exactly one of $p$ or $q$ is odd, note that every monomial in the expansion of $g(z)h(z)$ will have odd degree. However, the expectation of such monomials is always 0 as $z \in \{-1, +1\}^d$. $\qquad\square$

## B.2 Proof of Theorem 2.1

We now complete the proof of Theorem 2.1. Note that the kernel lower bound follows from B.1, so it suffices to upper bound the generalization error of the neural net solution.

*Proof of Theorem 2.1.* We first invoke Theorem C.2 to conclude that with $\lambda = \text{poly}(n)^{-1}$, the network $f^{\text{NN}}(\cdot; \Theta_\lambda)$ will have margin that is a constant factor approximation to the max-margin.

For neural nets with at least 4 hidden units, we now construct a neural net with a good normalized margin:

$$f^{\mathrm{NN}}(x) = [x^\top e_1]_+ + [-x^\top e_1]_+ - [x^\top e_2]_+ - [-x^\top e_2]_+$$

As this network has constant norm and margin 1, it has normalized margin $\Theta(1)$, and therefore the max neural net margin is $\Omega(1)$. Now we apply the generalization bound of Proposition D.1 to obtain

$$\Pr_{x,y \sim \mathcal{D}}[f^{\mathrm{NN}}(x; \Theta_\lambda)y \le 0] \lesssim \sqrt{\frac{d}{n}} + \sqrt{\frac{\log\log(16d)}{n}} + \sqrt{\frac{\log(1/\delta)}{n}}$$

as desired. Choosing $\delta = n^{-5}$ gives the desired result. Combined with the Theorem B.1 lower bound on the kernel method, this completes the proof. $\qquad\square$

### B.3 Regression Setting

In this section we argue that a analogue to Theorem 2.1 holds in the regression setting where we test on a truncated squared loss $\ell(\hat{y}; y) = \min((y - \hat{y})^2, 1)$. As the gap exists for the same distribution $\mathcal{D}$, the theorem statement is essentially identical to the classification setting, and the kernel lower bound carries over. For the regularized neural net upper bound, we will only highlight the differences here.

**Theorem B.10.** *Let $f^{\mathrm{NN}}(\cdot; \Theta)$ be some two-layer neural network with $m$ hidden units parametrized by $\Theta$, as in Section 2. Define the $\lambda$-regularized squared error loss*

$$L_{\lambda,m}(\Theta) \triangleq \frac{1}{n} \sum_{i=1}^{n} (f^{\mathrm{NN}}(x_i; \Theta) - y_i)^2 + \lambda \|\Theta\|_2^2$$

*with $\Theta_{\lambda,m} \in \arg\min_\Theta L_{\lambda,m}(\Theta)$. Suppose there exists a width-$m$ network that fits the data $(x_i, y_i)$ perfectly. Then as $\lambda \to 0$, $L_{\lambda,m}(\Theta_{\lambda,m}) \to 0$ and $\|\Theta_{\lambda,m}\|_2 \to \|\Theta^{\star,m}\|_2^2$, where $\Theta^{\star,m}$ is an optimizer of the following problem:*

$$\min_\Theta \|\Theta\|_2^2 \tag{B.14}$$
$$\text{such that } f^{\mathrm{NN}}(x_i; \Theta) = y_i \ \forall i$$

*Proof.* We note that $\lambda\|\Theta_{\lambda,m}\|_2^2 \le L_{\lambda,m}(\Theta_{\lambda,m}) \le L_{\lambda,m}(\Theta^{\star,m}) = \lambda\|\Theta^{\star,m}\|_2^2$, so as $\lambda \to 0$, and also $\|\Theta_{\lambda,m}\|_2 \le \|\Theta^{\star,m}\|_2$. Now assume for the sake of contradiction that $\exists B$ with $\|\Theta_{\lambda,m}\|_2 \le B < \|\Theta^{\star,m}\|_2$ for arbitrarily small $\lambda$. We define

$$r^\star \triangleq \min_\Theta \frac{1}{n} \sum_{i=1}^{n} (f^{\mathrm{NN}}(x_i; \Theta) - y_i)^2$$
$$\text{subject to } \|\Theta\|_2 \le B$$

Note that $r^\star > 0$ since $\Theta^{\star,m}$ is optimal for equation B.14. However, $L_{\lambda,m} \ge r^\star$ for arbitrarily small $\lambda$, a contradiction. Thus, $\lim_{\lambda \to 0} \|\Theta_{\lambda,m}\|_2^2 = \|\Theta^{\star,m}\|_2^2$. $\qquad\square$

For the distribution $\mathcal{D}$, the neural net from the proof of Theorem 2.1 also fits the data perfectly in the regression setting. As this network has norm $O(1)$, we can apply the norm-based Rademacher complexity bounds of Golowich et al. [25] in the same manner as in Section D (using standard tools for Lipschitz and bounded functions) to conclude a generalization error bound of $\tilde{O}\left(\sqrt{\frac{d\log n + \log(1/\delta)}{n}}\right)$, same as the classification upper bound.

### B.4 Connection to the $\ell_1$-SVM

In this section, we state a known connection between a $\ell_2$ regularized two-layer neural net and the $\ell_1$-SVM over relu features [48]. Following our notation from Section 4, we will use $\gamma^{\star,m}$ to denote the maximum possible normalized margin of a two-layer network with hidden layer size $m$ (note the emphasis on the size of the single hidden layer).

The depth $q = 2$ case of Corollary 4.2 implies that optimizing weakly-regularized $\ell_2$ loss over width-$m$ two-layer networks gives parameters whose generalization bounds depend on the hidden

layer size only through $1/\gamma^{\star,m}$. Furthermore, from Theorem 4.3 it immediately follows that $\gamma^{\star,1} \leq \gamma^{\star,2} \leq \cdots \leq \gamma^{\star,\infty}$. The work of Neyshabur et al. [48] links $\gamma^{\star,m}$ to the $\ell_1$ SVM over the lifted features $\varphi_{\text{relu}}$. We look at the margin of linear functionals corresponding to $\mu \in \mathcal{L}_1^1(\mathbb{S}^{d-1})$. The 1-norm SVM [75] over the lifted feature $\varphi_{\text{relu}}(x)$ solves for the maximum margin:

$$\gamma_{\ell_1} \triangleq \max_\mu \min_{i \in [n]} y_i \langle \mu, \varphi_{\text{relu}}(x_i) \rangle$$

$$\text{subject to } \|\mu\|_1 \leq 1 \tag{B.15}$$

This formulation is equivalent to a hard-margin optimization on "convex neural networks" [11]. Bach [7] also study optimization and generalization of convex neural networks. Using results from [60, 48, 11], our Theorem C.1 implies that optimizing weakly-regularized logistic loss over two-layer networks is equivalent to solving equation B.15 when the size of the hidden layer is at least $n + 1$. Proposition B.11 states this deduction.[6]

**Proposition B.11.** *Let $\gamma_{\ell_1}$ be defined in equation B.15. If margin $\gamma_{\ell_1}$ is attainable by some solution $\mu \in \mathcal{L}_1^1(\mathbb{S}^{d-1})$, then $\frac{\gamma_{\ell_1}}{2} = \gamma^{\star,n+1} = \cdots = \gamma^{\star,\infty}$.*

# C  Missing Material for Section 4

## C.1  Multi-class Setting

We will first state our analogue of Theorem 4.1 in the multi-class setting, as the proofs for the binary case will follow by reduction to the multi-class case.

In the same setting as Section 4, let $l$ be the number of multi-class labels, so the $i$-th example has label $y_i \in [l]$. Our family $\mathcal{F}$ of prediction functions $f$ now takes outputs in $\mathbb{R}^l$, and we now study the $\lambda$-regularized cross entropy loss, defined as

$$L_\lambda(\Theta) \triangleq -\frac{1}{n} \sum_{i=1}^n \log \frac{\exp(f_{y_i}(x_i; \Theta))}{\sum_{j=1}^l \exp(f_j(x_i; \Theta))} + \lambda \|\Theta\|^r \tag{C.1}$$

We redefine the normalized margin of $\Theta_\lambda$ as:

$$\gamma_\lambda \triangleq \min_i (f_{y_i}(x_i; \bar{\Theta}_\lambda) - \max_{j \neq y_i} f_j(x_i; \bar{\Theta}_\lambda)) \tag{C.2}$$

Define the $\|\cdot\|$-max normalized margin as

$$\gamma^\star \triangleq \max_{\|\Theta\| \leq 1} [\min_i (f_{y_i}(x_i; \Theta) - \max_{j \neq y_i} f_j(x_i; \Theta))]$$

and let $\Theta^\star$ be a parameter achieving this maximum. With these new definitions, our theorem statement for the multi-class setting is identical as the binary setting:

**Theorem C.1.** *Assume $\gamma^\star > 0$ in the multi-class setting with cross entropy loss. Then as $\lambda \to 0$, $\gamma_\lambda \to \gamma^\star$.*

Since $L_\lambda$ is typically hard to optimize exactly for neural nets, we study how accurately we need to optimize $L_\lambda$ to obtain a margin that approximates $\gamma^\star$ up to a constant. We show that for $\lambda$ polynomial in $n, \gamma^\star$, and $l$, it suffices to find $\Theta'$ achieving a constant factor $\alpha$ multiplicative approximation of $L_\lambda(\Theta_\lambda)$ in order to have margin $\gamma'$ satisfying $\gamma' \geq \frac{\gamma^\star}{\alpha^{a/r}}$.

**Theorem C.2.** *In the setting of Theorem C.1, suppose that we choose $\lambda = \exp(-(2^{r/a} - 1)^{-a/r}) \frac{(\gamma^\star)^{r/a}}{n^c(l-1)^c}$ for sufficiently large $c$ (that only depends on $r/a$). For $\alpha \leq 2$, let $\Theta'$ denote a $\alpha$-approximate minimizer of $L_\lambda$, so $L_\lambda(\Theta') \leq \alpha L_\lambda(\Theta_\lambda)$. Denote the normalized margin of $\Theta'$ by $\gamma'$. Then $\gamma' \geq \frac{\gamma^\star}{10 \cdot \alpha^{a/r}}$.*

Towards proving Theorem C.1, we first prove that $L_\lambda$ does indeed have a global minimizer.

**Claim C.3.** *In the setting of Theorems C.1 and 4.1, $\arg\min_\Theta L_\lambda(\Theta)$ exists.*

*Proof.* We will argue in the setting of Theorem C.1 where $L_\lambda$ is the multi-class cross entropy loss, because the logistic loss case is analogous. We first note that $L_\lambda$ is continuous in $\Theta$ because $f$ is continuous in $\Theta$ and the term inside the logarithm is always positive. Next, define $b \triangleq \inf_\Theta L_\lambda(\Theta) > 0$. Then we note that for $\|\Theta\| > (b/\lambda)^{1/r} \triangleq M$, we must have $L_\lambda(\Theta) > b$. It follows that $\inf_{\|\Theta\| \leq M} L_\lambda(\Theta) = \inf_\Theta L_\lambda(\Theta)$. However, there must be a value $\Theta_\lambda$ which attains $\inf_{\|\Theta\| \leq M} L_\lambda(\Theta)$, because $\{\Theta : \|\Theta\| \leq M\}$ is a compact set and $L_\lambda$ is continuous. Thus, $\inf_\Theta L_\lambda(\Theta)$ is attained by some $\Theta_\lambda$. $\qquad\square$

Next we present the following lemma, which says that as we decrease $\lambda$, the norm of the solution $\|\Theta_\lambda\|$ grows.

**Lemma C.4.** *In the setting of Theorem C.1, as $\lambda \to 0$, we have $\|\Theta_\lambda\| \to \infty$.*

To prove Theorem C.1, we rely on the exponential scaling of the cross entropy: $L_\lambda$ can be lower bounded roughly by $\exp(-\|\Theta_\lambda\|\gamma_\lambda)$, but also has an upper bound that scales with $\exp(-\|\Theta_\lambda\|\gamma^\star)$. By Lemma C.4, we can take large $\|\Theta_\lambda\|$ so the gap $\gamma^\star - \gamma_\lambda$ vanishes. This proof technique is inspired by that of Rosset et al. [58].

*Proof of Theorem C.1.* For any $M > 0$ and $\Theta$ with $\gamma_\Theta \triangleq \min_i \left( f(x_i; \bar{\Theta}) - \max_{j \neq y_i} f(x_i; \bar{\Theta}) \right)$,

$$L_\lambda(M\Theta) = \frac{1}{n} \sum_{i=1}^n - \log \frac{\exp(M^a f_{y_i}(x_i; \Theta))}{\sum_{j=1}^l \exp(M^a f_j(x_i; \Theta))} + \lambda M^r \|\Theta\|^r \quad \text{(by the homogeneity of } f\text{)}$$

$$= \frac{1}{n} \sum_{i=1}^n - \log \frac{1}{1 + \sum_{j \neq y_i} \exp(M^a(f_j(x_i; \Theta) - f_{y_i}(x_i; \Theta)))} + \lambda M^r \|\Theta\|^r \quad \text{(C.3)}$$

$$\leq \log(1 + (l-1)\exp(-M^a \gamma_\Theta)) + \lambda M^r \|\Theta\|^r \quad \text{(C.4)}$$

We can also apply $\sum_{j \neq y_i} \exp(M^a(f_j(x_i; \Theta) - f_{y_i}(x_i; \Theta))) \geq \max \exp(M^a(f_j(x_i; \Theta) - f_{y_i}(x_i; \Theta))) = \exp \gamma_\Theta$ in order to lower bound equation C.3 and obtain

$$L_\lambda(M\Theta) \geq \frac{1}{n} \log(1 + \exp(-M^a \gamma_\Theta)) + \lambda M^r \|\Theta\|^r \quad \text{(C.5)}$$

Applying equation C.4 with $M = \|\Theta_\lambda\|$ and $\Theta = \Theta^\star$, noting that $\|\Theta^\star\| \leq 1$, we have:

$$L_\lambda(\Theta^\star \|\Theta_\lambda\|) \leq \log(1 + (l-1)\exp(-\|\Theta_\lambda\|^a \gamma^\star)) + \lambda \|\Theta_\lambda\|^r \quad \text{(C.6)}$$

Next we lower bound $L_\lambda(\Theta_\lambda)$ by applying equation C.5,

$$L_\lambda(\Theta_\lambda) \geq \frac{1}{n} \log(1 + \exp(-\|\Theta_\lambda\|^a \gamma_\lambda)) + \lambda \|\Theta_\lambda\|^r \quad \text{(C.7)}$$

Combining equation C.6 and equation C.7 with the fact that $L_\lambda(\Theta_\lambda) \leq L_\lambda(\Theta^\star \|\Theta_\lambda\|)$ (by the global optimality of $\Theta_\lambda$), we have

$$\forall \lambda > 0, n \log(1 + (l-1)\exp(-\|\Theta_\lambda\|^a \gamma^\star)) \geq \log(1 + \exp(-\|\Theta_\lambda\|^a \gamma_\lambda))$$

Recall that by Lemma C.4, as $\lambda \to 0$, we have $\|\Theta_\lambda\| \to \infty$. Therefore, $\exp(-\|\Theta_\lambda\|^a \gamma^\star), \exp(-\|\Theta_\lambda\|^a \gamma_\lambda) \to 0$. Thus, we can apply Taylor expansion to the equation above with respect to $\exp(-\|\Theta_\lambda\|^a \gamma^\star)$ and $\exp(-\|\Theta_\lambda\|^a \gamma_\lambda)$. If $\max\{\exp(-\|\Theta_\lambda\|^a \gamma^\star), \exp(-\|\Theta_\lambda\|^a \gamma_\lambda)\} < 1$, then we obtain

$$n(l-1)\exp(-\|\Theta_\lambda\|^a \gamma^\star) \geq \exp(-\|\Theta_\lambda\|^a \gamma_\lambda) - O(\max\{\exp(-\|\Theta_\lambda\|^a \gamma^\star)^2, \exp(-\|\Theta_\lambda\|^a \gamma_\lambda)^2\})$$

We claim this implies that $\gamma^\star \leq \liminf_{\lambda \to 0} \gamma_\lambda$. If not, we have $\liminf_{\lambda \to 0} \gamma_\lambda < \gamma^\star$, which implies that the equation above is violated with sufficiently large $\|\Theta_\lambda\|$ ($\|\Theta_\lambda\| \gg \log(2(\ell-1)n)^{1/a}$ would suffice). By Lemma C.4, $\|\Theta_\lambda\| \to \infty$ as $\lambda \to 0$ and therefore we get a contradiction.

Finally, we have $\gamma_\lambda \leq \gamma^\star$ by definition of $\gamma^\star$. Hence, $\lim_{\lambda \to 0} \gamma_\lambda$ exists and equals $\gamma^\star$. $\qquad\square$

Now we fill in the proof of Lemma C.4.

*Proof of Lemma C.4.* For the sake of contradiction, we assume that $\exists C > 0$ such that for any $\lambda_0 > 0$, there exists $0 < \lambda < \lambda_0$ with $\|\Theta_\lambda\| \leq C$. We will determine the choice of $\lambda_0$ later and pick $\lambda$ such that $\|\Theta_\lambda\| \leq C$. Then the logits (the prediction $f_j(x_i; \Theta)$ before softmax) are bounded in absolute value by some constant (that depends on $C$), and therefore the loss function $-\log \frac{\exp(f_{y_i}(x_i;\Theta))}{\sum_{j=1}^l \exp(f_j(x_i;\Theta))}$ for every example is bounded from below by some constant $D > 0$ (depending on $C$ but not $\lambda$.)

Let $M = \lambda^{-1/(r+1)}$, we have that

$$0 < D \leq L_\lambda(\Theta_\lambda) \leq L_\lambda(M\Theta^\star) \qquad \text{(by the optimality of } \Theta_\lambda)$$

$$\leq -\log \frac{1}{1 + (l-1)\exp(-M^a \gamma^\star)} + \lambda M^r \qquad \text{(by equation C.4)}$$

$$= \log(1 + (l-1)\exp(-\lambda^{-a/(r+1)}\gamma^\star)) + \lambda^{1/(r+1)}$$

$$\leq \log(1 + (l-1)\exp(-\lambda_0^{-a/(r+1)}\gamma^\star)) + \lambda_0^{1/(r+1)}$$

Taking a sufficiently small $\lambda_0$, we obtain a contradiction and complete the proof. $\square$

## C.2  Missing Proof for Optimization Accuracy

*Proof of Theorem C.2.* Choose $B \triangleq \left(\frac{1}{\gamma^\star}\log\frac{(l-1)(\gamma^\star)^{r/a}}{\lambda}\right)^{1/a}$. We can upper bound $L_\lambda(\Theta')$ by computing

$$L_\lambda(\Theta') \leq \alpha L\lambda(\Theta_\lambda) \leq \alpha L_\lambda(B\Theta^\star)$$

$$\leq \alpha \log(1 + (l-1)\exp(-B^a\gamma^\star)) + \alpha\lambda B^r \qquad \text{(by equation C.4)}$$

$$\leq \alpha(l-1)\exp(-B^a\gamma^\star) + \alpha\lambda B^r \qquad \text{(using } \log(1+x) \leq x)$$

$$\leq \alpha\frac{\lambda}{(\gamma^\star)^{r/a}} + \alpha\lambda\left(\frac{1}{\gamma^\star}\log\frac{(l-1)(\gamma^\star)^{r/a}}{\lambda}\right)^{r/a}$$

$$\leq \alpha\frac{\lambda}{(\gamma^\star)^{r/a}}\left(1 + \left(\log\frac{(l-1)(\gamma^\star)^{r/a}}{\lambda}\right)^{r/a}\right) \triangleq L^{(UB)}$$

Furthermore, it holds that $\|\Theta'\|^r \leq \frac{L^{(UB)}}{\lambda}$. Now we note that

$$L_\lambda(\Theta') \leq L^{(UB)} \leq 2\alpha\frac{\lambda}{(\gamma^\star)^{r/a}}\left(\log\frac{(l-1)(\gamma^\star)^{r/a}}{\lambda}\right)^{r/a} \leq \frac{1}{2n}$$

for sufficiently large $c$ depending only on $a/r$. Now using the fact that $\log(x) \geq \frac{x}{1+x} \, \forall x \geq -1$, we additionally have the lower bound $L_\lambda(\Theta') \geq \frac{1}{n}\log(1 + \exp(-\gamma'\|\Theta'\|^a)) \geq \frac{1}{n}\frac{\exp(-\gamma'\|\Theta'\|^a)}{1+\exp(-\gamma'\|\Theta'\|^a)}$. Since $L^{(UB)} \leq 1$, we can rearrange to get

$$\gamma' \geq \frac{-\log\frac{nL_\lambda(\Theta')}{1-nL_\lambda(\Theta')}}{\|\Theta'\|^a} \geq \frac{-\log\frac{nL^{(UB)}}{1-nL^{(UB)}}}{\|\Theta'\|^a} \geq \frac{-\log(2nL^{(UB)})}{\|\Theta'\|^a}$$

The middle inequality followed because $\frac{x}{1-x}$ is increasing in $x$ for $0 \leq x < 1$, and the last because $L^{(UB)} \leq \frac{1}{2n}$. Since $-\log 2nL^{(UB)} > 0$ we can also apply the bound $\|\Theta'\|^r \leq \frac{L^{(UB)}}{\lambda}$ to get

$$
\gamma' \geq \frac{-\lambda^{a/r} \log 2nL^{(UB)}}{(L^{(UB)})^{a/r}}
$$

$$
= \frac{-\log\left(2n\alpha\frac{\lambda}{(\gamma^\star)^{r/a}}\left(1 + \left(\log\frac{(l-1)(\gamma^\star)^{r/a}}{\lambda}\right)^{r/a}\right)\right)}{\frac{\alpha^{a/r}}{\gamma^\star}\left(1 + \left(\log\frac{(l-1)(\gamma^\star)^{r/a}}{\lambda}\right)^{r/a}\right)^{a/r}} \qquad \text{(by definition of } L^{(UB)})
$$

$$
\geq \frac{\gamma^\star}{\alpha^{a/r}} \left( \underbrace{\frac{\log(\frac{(\gamma^\star)^{r/a}}{2\alpha n\lambda})}{\left(1 + \left(\log\frac{(l-1)(\gamma^\star)^{r/a}}{\lambda}\right)^{r/a}\right)^{a/r}}}_{\clubsuit} - \underbrace{\frac{\log\left(1 + \left(\log\frac{(l-1)(\gamma^\star)^{r/a}}{\lambda}\right)^{r/a}\right)}{\left(1 + \left(\log\frac{(l-1)(\gamma^\star)^{r/a}}{\lambda}\right)^{r/a}\right)^{a/r}}}_{\heartsuit} \right)
$$

We will first bound $\clubsuit$. First note that

$$
\frac{\log(\frac{(\gamma^\star)^{r/a}}{2\alpha n\lambda})}{\log\frac{(l-1)(\gamma^\star)^{r/a}}{\lambda}} = \frac{\log\frac{(\gamma^\star)^{r/a}}{\lambda} - \log 2\alpha n}{\log\frac{(\gamma^\star)^{r/a}}{\lambda} + \log(l-1)} \geq \frac{\log\frac{(\gamma^\star)^{r/a}}{\lambda} - \log 2\alpha n(l-1)}{\log\frac{(\gamma^\star)^{r/a}}{\lambda}} \geq \frac{c-3}{c} \qquad \text{(C.8)}
$$

where the last inequality follows from the fact that $\frac{(\gamma^\star)^{r/a}}{\lambda} \geq n^c(l-1)^c$ and $\alpha \leq 2$. Next, using the fact that $\log\frac{(\gamma^\star)^{r/a}}{\lambda} \geq \frac{1}{(2^{r/a}-1)^{a/r}}$, we note that

$$
\left(1 + \left(\log\frac{(l-1)(\gamma^\star)^{r/a}}{\lambda}\right)^{-r/a}\right)^{a/r} \leq \left(1 + \left(\frac{1}{(2^{r/a}-1)^{a/r}}\right)^{-r/a}\right)^{a/r} \leq 2 \qquad \text{(C.9)}
$$

Combining equation C.8 and equation C.9, we can conclude that

$$
\clubsuit = \frac{\log(\frac{(\gamma^\star)^{r/a}}{2\alpha n\lambda})}{\log\frac{(l-1)(\gamma^\star)^{r/a}}{\lambda}} \left(1 + \left(\log\frac{(l-1)(\gamma^\star)^{r/a}}{\lambda}\right)^{-r/a}\right)^{-a/r} \geq \frac{c-3}{2c}
$$

Finally, we note that if $1 + \left(\log\frac{(l-1)(\gamma^\star)^{r/a}}{\lambda}\right)^{r/a}$ is a sufficiently large constant that depends only on $a/r$ (which can be achieved by choosing $c$ sufficiently large) it will follow that $\heartsuit \leq \frac{1}{10}$. Thus, for sufficiently large $c \geq 5$, we can combine our bounds on $\clubsuit$ and $\heartsuit$ to get that

$$
\gamma' \geq \frac{\gamma^\star}{10\alpha^{a/r}}
$$

$\square$

## C.3 Proofs of Theorem 4.1

For completeness, we will now prove Theorem 4.1 via reduction to the multi-class cases. Recall that we now fit binary labels $y_i \in \{-1, +1\}$ (as opposed to indices in $[l]$) and redefine $f(\cdot; \Theta)$ to assign a single real-valued score (as opposed to a score for each label). We also work with the simpler logistic loss in equation 4.1.

*Proof of Theorem 4.1.* We prove this theorem via reduction to the multi-class case with $l = 2$. Construct $\tilde{f} : \mathbb{R}^d \to \mathbb{R}^2$ with $\tilde{f}_1(x_i; \Theta) = -\frac{1}{2}f(x_i; \Theta)$ and $\tilde{f}_2(x_i; \Theta) = \frac{1}{2}f(x_i; \Theta)$. Define new labels $\tilde{y}_i = 1$ if $y_i = -1$ and $\tilde{y}_i = 2$ if $y_i = 1$. Now note that $\tilde{f}_{\tilde{y}_i}(x_i; \Theta) - \tilde{f}_{j\neq\tilde{y}_i}(x_i; \Theta) = y_i f(x_i; \Theta)$,

so the multi-class margin for $\Theta$ under $\tilde{f}$ is the same as binary margin for $\Theta$ under $f$. Furthermore, defining

$$\tilde{L}_\lambda(\Theta) \triangleq \frac{1}{n}\sum_{i=1}^n - \log \frac{\exp(\tilde{f}_{\tilde{y}_i}(x_i;\Theta))}{\sum_{j=1}^2 \exp(\tilde{f}_j(x_i;\Theta))} + \lambda\|\Theta\|^r$$

we get that $\tilde{L}_\lambda(\Theta) = L_\lambda(\Theta)$, and in particular, $\tilde{L}_\lambda$ and $L_\lambda$ have the same set of minimizers. Therefore we can apply Theorem C.1 for the multi-class setting and conclude $\gamma_\lambda \to \gamma^\star$ in the binary classification setting. $\qquad\square$

## D Generalization Bounds for Neural Nets

In this section we present generalization bounds in terms of the normalized margin and complete the proof of Corollary 4.2. We first state the following Proposition D.1, which shows that the generalization error only depends on the parameters through the inverse of the margin on the training data. We obtain Proposition D.1 by applying Theorem 1 of Golowich et al. [25] with the standard technique of using margin loss to bound classification error. There exist other generalization bounds which depend on the margin and some normalization [50, 51, 9, 53]; we choose the bounds of Golowich et al. [25] because they fit well with $\ell_2$ normalization.

**Proposition D.1.** *[Straightforward consequence of Golowich et al. [25, Theorem 1]] Suppose $\phi$ is 1-Lipschitz and 1-positive-homogeneous. With probability at least $1 - \delta$ over the draw of $X, Y$, for all depth-$q$ networks $f^{\mathrm{NN}}(\cdot; \Theta)$ separating the data with normalized margin $\gamma \triangleq \min_i y_i f^{\mathrm{NN}}(x_i; \Theta/\|\Theta\|_F) > 0$,*

$$L(\Theta) \lesssim \frac{C}{\gamma q^{(q-1)/2}\sqrt{n}} + \epsilon(\gamma) \tag{D.1}$$

*where $\epsilon(\gamma) \triangleq \sqrt{\frac{\log\log_2 \frac{4C}{\gamma}}{n}} + \sqrt{\frac{\log(1/\delta)}{n}}$ and $C = \max_{x\in\mathcal{X}}\|x\|_2$ is the max norm of the data. Note that $\epsilon(\gamma)$ is typically small, and thus the above bound mainly scales with $\frac{C}{\gamma q^{(q-1)/2}\sqrt{n}}$.* [7]

We note that Proposition D.1 is stated directly in terms of the normalized margin in order to maintain consistency in our notation, whereas prior works state their results using a ratio between unnormalized margin and norms of the weight matrices [9]. We provide the proof in the following section.

### D.1 Proof of Proposition D.1

We prove the generalization error bounds stated in Proposition D.1 via Rademacher complexity and margin theory.

Assume that our data $X, Y$ are drawn i.i.d. from ground truth distribution $p_{\mathrm{data}}$ supported on $\mathcal{X} \times \mathcal{Y}$. For some hypothesis class $\mathcal{F}$ of real-valued functions, we define the empirical Rademacher complexity $\hat{\mathfrak{R}}(\mathcal{F})$ as follows:

$$\hat{\mathfrak{R}}(\mathcal{F}) \triangleq \frac{1}{n}\mathbb{E}_{\epsilon_i}\left[\sup_{f\in\mathcal{F}}\sum_{i=1}^n \epsilon_i f(x_i)\right]$$

where $\epsilon_i$ are independent Rademacher random variables. For a classifier $f$, following the notation of Section 4.1 we will use $L(f) \triangleq \Pr_{(x,y)\sim p_{\mathrm{data}}}(yf(x) \le 0)$ to denote the population 0-1 loss of the classifier $f$. The following classical theorem [34], [33] bounds generalization error in terms of the Rademacher complexity and margin loss.

**Theorem D.2** (Theorem 2 of Kakade et al. [33])**.** *Let $(x_i, y_i)_{i=1}^n$ be drawn iid from $p_{\mathrm{data}}$. We work in the binary classification setting, so $\mathcal{Y} = \{-1, 1\}$. Assume that for all $f \in \mathcal{F}$, we have $\sup_{x\in\mathcal{X}}|f(x)| \le C$. Then with probability at least $1 - \delta$ over the random draws of the data, for every $\gamma > 0$ and $f \in \mathcal{F}$,*

$$L(f) \le \frac{1}{n}\sum_{i=1}^n \mathbb{1}(y_i f(x_i) < \gamma) + \frac{4\hat{\mathfrak{R}}(\mathcal{F})}{\gamma} + \sqrt{\frac{\log\log_2 \frac{4C}{\gamma}}{n}} + \sqrt{\frac{\log(1/\delta)}{2n}}$$

We will prove Proposition D.1 by applying the Rademacher complexity bounds of Golowich et al. [25] with Theorem D.2.

First, we show the following lemma bounding the generalization of neural networks whose weight matrices have bounded Frobenius norms. For this proof we drop the superscript $^{\mathrm{NN}}$ as it is clear from context.

**Lemma D.3.** *Define the hypothesis class $\mathcal{F}_q$ over depth-$q$ neural networks by*

$$\mathcal{F}_q = \left\{ f(\cdot; \Theta) : \|W_j\|_F \leq \frac{1}{\sqrt{q}} \ \forall j \right\}$$

*Let $C \triangleq \sup_{x \in \mathcal{X}} \|x\|_2$. Recall that $L(\Theta)$ denotes the 0-1 population loss $L(f(\cdot; \Theta))$. Then for any $f(\cdot; \Theta) \in \mathcal{F}_q$ classifying the training data correctly with unnormalized margin $\gamma_\Theta \triangleq \min_i y_i f(x_i; \Theta) > 0$, with probability at least $1 - \delta$,*

$$L(\Theta) \lesssim \frac{C}{\gamma_\Theta q^{(q-1)/2} \sqrt{n}} + \sqrt{\frac{\log \log_2 \frac{4C}{\gamma_\Theta}}{n}} + \sqrt{\frac{\log(1/\delta)}{n}} \tag{D.2}$$

*Note the dependence on the unnormalized margin rather than the normalized margin.*

*Proof.* We first claim that $\sup_{f(\cdot; \Theta) \in \mathcal{F}_q} \sup_{x \in \mathcal{X}} f(x; \Theta) \leq C$. To see this, for any $f(\cdot; \Theta) \in \mathcal{F}_q$,

$$
\begin{aligned}
f(x; \Theta) &= W_q \phi(\cdots \phi(W_1 x) \cdots) \\
&\leq \|W_q\|_F \|\phi(W_{q-1} \phi(\cdots \phi(W_1 x) \cdots))\|_2 \\
&\leq \|W_q\|_F \|W_{q-1} \phi(\cdots \phi(W_1 x) \cdots)\|_2 \\
&\qquad \text{(since } \phi \text{ is 1-Lipschitz and } \phi(0) = 0 \text{, so } \phi \text{ performs a contraction)} \\
&< \|x\|_2 \leq C \qquad \text{(repeatedly applying this argument and using } \|W_j\|_F < 1)
\end{aligned}
$$

Furthermore, by Theorem 1 of Golowich et al. [25], $\hat{\mathfrak{R}}(\mathcal{F}_q)$ has upper bound

$$\hat{\mathfrak{R}}(\mathcal{F}_q) \lesssim \frac{C}{q^{(q-1)/2} \sqrt{n}}$$

Thus, we can apply Theorem D.2 to conclude that for all $f(\cdot; \Theta) \in \mathcal{F}_q$ and all $\gamma > 0$, with probability $1 - \delta$,

$$L(\Theta) \lesssim \frac{1}{n} \sum_{i=1}^n \mathbb{1}(y_i f(x_i; \Theta) < \gamma) + \frac{C}{\gamma q^{(q-1)/2} \sqrt{n}} + \sqrt{\frac{\log \log_2 \frac{4C}{\gamma}}{n}} + \sqrt{\frac{\log(1/\delta)}{n}}$$

In particular, by definition choosing $\gamma = \gamma_\Theta$ makes the first term on the LHS vanish and gives the statement of the lemma. $\qquad \square$

*Proof of Proposition D.1.* Given parameters $\Theta = (W_1, \ldots, W_q)$, we first construct parameters $\tilde{\Theta} = (\tilde{W}_1, \ldots, \tilde{W}_q)$ such that $f(\cdot; \bar{\Theta})$ and $f(\cdot; \tilde{\Theta})$ compute the same function, and $\|\tilde{W}_1\|_F^2 = \|\tilde{W}_2\|_F^2 = \cdots = \|\tilde{W}_q\|_F^2 \leq \frac{1}{q}$. To do this, we set

$$\tilde{W}_j = \frac{(\prod_{k=1}^q \|W_k\|_F)^{1/k}}{\|W_j\|_F \|\Theta\|_F} W_j$$

By construction

$$
\begin{aligned}
\|\tilde{W}_j\|_F^2 &= \frac{(\prod_{k=1}^q \|W_k\|_F^2)^{1/k}}{\|\Theta\|_F^2} \\
&= \frac{(\prod_{k=1}^q \|W_k\|_F^2)^{1/k}}{\sum_{k=1}^q \|W_k\|_F^2} \\
&\leq \frac{1}{k} \qquad \text{(by the AM-GM inequality)}
\end{aligned}
$$

Furthermore, we also have

$$\begin{aligned}
f(x; \tilde{\Theta}) &= \tilde{W}_q \phi(\cdots \phi(\tilde{W}_1 x) \cdots) \\
&= \prod_{j=1}^{q} \frac{(\prod_{k=1}^{q} \|W_k\|_F)^{1/k}}{\|W_j\|_F \|\Theta\|_F} W_q \phi(\cdots \phi(W_1 x) \cdots) \qquad \text{(by the homogeneity of } \phi) \\
&= \frac{1}{\|\Theta\|_F^q} f(x; \Theta) \\
&= f\left(x; \frac{\Theta}{\|\Theta\|_F}\right) \qquad \text{(since } f \text{ is } q\text{-homogeneous in } \Theta) \\
&= f(x; \bar{\Theta})
\end{aligned}$$

Now we note that by construction, $L(\Theta) = L(\tilde{\Theta})$. Now $f(\cdot; \tilde{\Theta})$ must also classify the training data perfectly, has unnormalized margin $\gamma$, and furthermore $f(\cdot; \tilde{\Theta}) \in \mathcal{F}_q$. As a result, Lemma D.3 allows us to conclude the desired statement. $\qquad\square$

To conclude Corollary 4.2, we apply the above on $\Theta_{\lambda,\mathcal{M}}$ and use Theorem 4.1.

*Proof of Corollary 4.2.* Applying the statement of Proposition D.1, with probability $1 - \delta$, for all $\lambda > 0$,

$$L(\Theta_{\lambda,\mathcal{M}}) \lesssim \frac{C}{\gamma_{\lambda,\mathcal{M}} q^{(q-1)/2} \sqrt{n}} + \epsilon(\gamma_{\lambda,\mathcal{M}})$$

Now we take the $\limsup$ of both sides as $\lambda \to 0$:

$$\begin{aligned}
\limsup_{\lambda \to 0} L(\Theta_{\lambda,\mathcal{M}}) &\lesssim \limsup_{\lambda \to 0} \frac{C}{\gamma_{\lambda,\mathcal{M}} q^{(q-1)/2} \sqrt{n}} + \epsilon(\gamma_{\lambda,\mathcal{M}}) \\
&\lesssim \frac{C}{\gamma^{\star,\mathcal{M}} q^{(q-1)/2} \sqrt{n}} + \epsilon(\gamma^{\star,\mathcal{M}}) \qquad \text{(by Theorem 4.1)}
\end{aligned}$$

$\qquad\square$

# E  Missing Proofs in Section 3

## E.1  Detailed Setup

We first write our regularity assumptions on $\Phi$, $R$, and $V$ in more detail:

**Assumption E.1** (Regularity conditions on $\Phi$, $R$, $V$). *$R$ is convex, nonnegative, Lipschitz, and smooth: $\exists M_R, C_R$ such that $\|\nabla^2 R\|_{op} \leq C_R$, and $\|\nabla R\|_2 \leq M_R$.*

**Assumption E.2.** *$\Phi$ is differentiable, bounded and Lipschitz on the sphere: $\exists B_\Phi, M_\Phi$ such that $\|\Phi(\bar{\theta})\| \leq B_\Phi \ \forall \bar{\theta} \in \mathbb{S}^d$, and $|\Phi_i(\bar{\theta}) - \Phi_i(\bar{\theta}')| \leq M_\Phi \|\bar{\theta} - \bar{\theta}'\|_2 \ \forall \bar{\theta}, \bar{\theta}' \in \mathbb{S}^d$.*

**Assumption E.3.** *$V$ is Lipschitz and upper and lower bounded on the sphere: $\exists b_V, B_V, M_V$ such that $0 < b_V \leq V(\bar{\theta}) \leq B_V \ \forall \bar{\theta} \in \mathbb{S}^d$, and $\|\nabla V(\bar{\theta})\|_2 \leq M_V \ \forall \bar{\theta} \in \mathbb{S}^d$.*

We state the version of Theorem 3.3 that collects these parameters:

**Theorem E.4** (Theorem 3.3 with problem parameters). *Suppose that $\Phi$ and $V$ are 2-homogeneous and Assumptions E.1, E.2, and E.3 hold. Fix a desired error threshold $\epsilon > 0$. Suppose that from a starting distribution $\rho_0$, a solution to the dynamics in equation 3.2 exists. Choose*

$$\sigma \triangleq \exp(-d \log(1/\epsilon) \mathrm{poly}(k, M_V, M_R, M_\Phi, b_V, B_V, C_R, B_\Phi, L[\rho_0] - L^\star))$$

$$t_\epsilon \triangleq \frac{d^2}{\epsilon^4} \mathrm{poly}(\log(1/\epsilon), k, M_V, M_R, M_\Phi, b_V, B_V, C_R, B_\Phi, L[\rho_0] - L^\star)$$

*Then it must hold that $\min_{0 \leq t \leq t_\epsilon} L[\rho_t] - \inf_\rho L[\rho] \leq 2\epsilon$.*

## E.2  Proof Outline of Theorem E.4

In this section, we will provide an outline of the proof of Theorem E.4. We will fill in the missing details in Section E.3.

Throughout the proof, it will be useful to keep track of $W_t \triangleq \sqrt{\mathbb{E}_{\theta \sim \rho_t}[\|\theta\|_2^2]}$, which measures the second moment of $\rho_t$. For convenience, we will also define the constant $B_L \triangleq M_R B_\Phi + B_V$. The following lemma first states that this second moment will never become too large.

**Lemma E.5.** *Choose any $t \leq \sigma B_L / b_V$. For all $0 \leq t' \leq t$, $W_{t'}^2 \leq \frac{L[\rho_0] + \sigma t B_L}{b_V - t \sigma B_L}$. In particular, for all $t \leq t_\epsilon$, we have $W_t \leq W_\epsilon$, where $W_\epsilon$ is defined as follows:*

$$W_\epsilon \triangleq \sqrt{\frac{L[\rho_0] + \sigma t_\epsilon B_L}{b_v - t_\epsilon \sigma B_L}} \tag{E.1}$$

Next, we will prove the following statement, which intuitively says that for an arbitrary choice of $\bar{\theta} \in \mathbb{S}^d$, if $L'[\rho_t](\bar{\theta})$ changes by a large amount between time steps $t$ and $t + l$, the objective function must also have decreased a lot.

**Lemma E.6.** *Define the quantity $Q(t) \triangleq \int \Phi d\rho_t$. For every $\bar{\theta} \in \mathbb{S}^d$ and $0 \leq t \leq t + l \leq t_\epsilon$, $\exists c_1 \triangleq \mathrm{poly}(k, C_R, B_\Phi, M_\Phi, B_L)$ such that*

$$|L'[\rho_t](\bar{\theta}) - L'[\rho_{t+l}](\bar{\theta})| \leq C_R B_\Phi \int_t^{t+l} \|Q'(t)\|_1 \tag{E.2}$$

$$\leq \sigma l c_1 (W_\epsilon^2 + 1) + c_1 W_\epsilon \sqrt{l} (L[\rho_t] - L[\rho_{t+l}] + \sigma l c_1 (W_\epsilon^2 + 1))^{1/2} \tag{E.3}$$

*where $W_\epsilon$ is defined as in equation E.1.*

The proof of Lemma E.6 intuitively holds because in order for $L'[\rho_t](\bar{\theta})$ to change by a large amount, the gradient flow dynamics must have shifted $\rho_t$ by some amount, which would have resulted in some decrease of the objective $L[\rho_t]$. We will rely on the 2-homogeneity of $\Phi$ to formalize this argument.

Next, we will rely on the convexity of $L$: letting $\rho^\star$ be an $\epsilon$-approximate global optimizer of $L$, since $L$ is convex in $\rho$, we have

$$L[\rho^\star] \geq L[\rho_t] + \mathbb{E}_{\theta \sim \rho^\star}[L'[\rho_t](\theta)] - \mathbb{E}_{\theta \sim \rho_t}[L'[\rho_t](\theta)]$$

Thus, if $\rho_t$ is far from optimality, it follows that either 1) the quantity $\mathbb{E}_{\theta \sim \rho_t}[L'[\rho_t](\theta)]$ has a large positive value or 2) there exists some descent direction $\bar{\theta} \in \mathbb{S}^d$ for which $L'[\rho_t](\theta) \ll 0$.

For the first case, we have the following guarantee that the objective decreases by a large amount:

**Lemma E.7.** *For any time $t$ with $0 \leq t \leq t_\epsilon$, we have*

$$\frac{d}{dt} L[\rho_t] \leq \sigma B_L (W_\epsilon^2 + 1) - \frac{\mathbb{E}_{\theta \sim \rho_t}[L'[\rho_t](\theta)]^2}{W_\epsilon^2} \tag{E.4}$$

Lemma E.7 relies on the 2-homogeneity of $\Phi$ and $R$ and is proven via arguing that the gradient flow dynamics will result in a large shift in $\rho_t$ and therefore substantial decrease in loss.

For the second case, we will show that the $\sigma U^d$ noise term will cause mass to grow exponentially fast in this descent direction until we make progress in decreasing the objective.

**Lemma E.8.** *Fix any $\tau > 0$. Choose time interval length $l$ by*

$$l \geq \frac{\log(W_\epsilon^2 / \sigma) + 2d \log \frac{2c_2}{\tau}}{\tau - \sigma} + 1$$

*If $\exists \bar{\theta} \in \mathbb{S}^d$ with $L'[\rho_{t^*}](\bar{\theta}) \leq -\tau$ for some $t^*$ satisfying $t^* + l \leq t_\epsilon$, then after $l$ steps, we will have*

$$L[\rho_{t^*+l}] \leq L[\rho_{t^*}] - \frac{(\tau/4 - \sigma l c_1 (W_\epsilon^2 + 1))^2}{l c_1^2 W_\epsilon^2} + \sigma l c_1 (W_\epsilon^2 + 1) \tag{E.5}$$

*Here $c_1$ is the constant defined in Lemma E.6 and $c_2$ is defined by $c_2 \triangleq \sqrt{k} M_R M_\Phi + M_V$.*

Lemma E.8 is proven via the following argument: first, if $L'[\rho_t](\bar{\theta})$ is close to $-\tau$ for all $t \in [t^*, t^*+l]$, then from the 2-homogeneity of $\Phi$ and $R$, the mass of $\rho_t$ in the neighborhood around $\bar{\theta}$ will grow exponentially fast, leading to a violation of Lemma E.5. (Because of the uniform noise injected into the gradient flow dynamics, $\rho_t$ will always have some mass in the neighborhood of $\bar{\theta}$ to start with.) Thus, it follows that $L'[\rho_t](\bar{\theta})$ must change by at least $\tau/4$, allowing us to invoke Lemma E.6 to argue that the objective must drop.

Lemmas E.7 and E.8 are enough to ensure that the objective will always decrease a sufficient amount after some polynomial-size time interval. This allows us to complete the proof of Theorem E.4 below:

*Proof of Theorem E.4.* Let $L^\star$ denote the infimum $\inf_\rho L[\rho]$, and let $\rho^\star$ be an $\epsilon$-approximate global minimizer of $L$: $L[\rho^\star] \leq L^\star + \epsilon$. (We define $\rho^\star$ because a true minimizer of $L$ might not exist.) Let $W^\star \triangleq \mathbb{E}_{\theta \sim \rho^\star}[\|\theta\|_2^2]$. We first note that since $b_V W^{\star 2} \leq L[\rho^\star] \leq L[\rho_0]$, $W^{\star 2} \leq L[\rho_0]/b_V \leq W_\epsilon^2$.

Now we bound the suboptimality of $\rho_t$: since $L$ is convex in $\rho$,

$$L[\rho^\star] \geq L[\rho_t] + \mathbb{E}_{\theta \sim \rho^\star}[L'[\rho_t](\theta)] - \mathbb{E}_{\theta \sim \rho_t}[L'[\rho_t](\theta)]$$

Rearranging gives

$$L[\rho_t] - L[\rho^\star] \leq \mathbb{E}_{\theta \sim \rho_t}[L'[\rho_t](\theta)] - \mathbb{E}_{\theta \sim \rho^\star}[L'[\rho_t](\theta)]$$

$$\leq \mathbb{E}_{\theta \sim \rho_t}[L'[\rho_t](\theta)] - W^{\star 2} \min\left\{\min_{\bar{\theta} \in \mathbb{S}^{d-1}} L'[\rho_t](\bar{\theta}), 0\right\} \tag{E.6}$$

Now let $l \triangleq \frac{W_\epsilon^2}{\epsilon - 2W_\epsilon^2 \sigma}\left(2\log\frac{W_\epsilon^2}{\sigma} + 2d\log\frac{4W_\epsilon^2 c_2}{\epsilon}\right)$, which satisfies Lemma E.8 with the value of $\tau$ later specified. Suppose that there is a $t$ with $0 \leq t \leq t_\epsilon - 2l$ and $\forall t' \in [t, t+2l]$, $L[\rho_{t'}] - L^\star \geq 2\epsilon$. Then $L[\rho_{t'}] - L[\rho^\star] \geq \epsilon$. We will argue that the objective decreases when we are $\epsilon$ suboptimal:

$$L[\rho_t] - L[\rho_{t+2l}] \geq \tag{E.7}$$
$$\min\left\{\frac{(\epsilon/8W_\epsilon^2 - l\sigma c_1(W_\epsilon^2 + 1))^2}{c_1^2 W_\epsilon^2 l} - 3\sigma l c_1(W_\epsilon^2 + 1), l\frac{\epsilon^2}{4W_\epsilon^2} - 2\sigma l B_L(W_\epsilon^2 + 1)\right\} \tag{E.8}$$

Using equation E.6 and $W_\epsilon \geq W^\star$, we first note that

$$\epsilon \leq \mathbb{E}_{\theta \sim \rho_{t'}}[L'[\rho_{t'}](\theta)] - W_\epsilon^2 \min\left\{\min_{\bar{\theta} \in \mathbb{S}^{d-1}} L'[\rho_{t'}](\bar{\theta}), 0\right\} \quad \forall t' \in [t, t+l]$$

Thus, either $\min_{\bar{\theta} \in \mathbb{S}^d} L'[\rho_{t'}](\bar{\theta}) \leq -\frac{\epsilon}{2W^{\star 2}} \leq -\frac{\epsilon}{2W_\epsilon^2}$, or $\mathbb{E}_{\theta \sim \rho_{t'}}[L'[\rho_{t'}](\theta)] \geq \frac{\epsilon}{2}$. If $\exists t' \in [t, t+l]$ such that the former holds, then we can apply Lemma E.8 with $\tau \triangleq \frac{\epsilon}{2W_\epsilon^2}$ to obtain

$$L[\rho_{t'}] - L[\rho_{t'+l}] \geq \frac{(\epsilon/8W_\epsilon^2 - l\sigma c_1(W_\epsilon^2 + 1))^2}{c_1^2 W_\epsilon^2 l} - \sigma l c_1(W_\epsilon^2 + 1)$$

Furthermore, from Lemma E.13, $L[\rho_{t+2l}] - L[\rho_{t'+l}] \leq \sigma l c_1(W_\epsilon^2 + 1)$ and $L[\rho_{t'}] - L[\rho_t] \leq \sigma l B_L(W_\epsilon^2 + 1)$, and so combining gives

$$L[\rho_t] - L[\rho_{t+2l}] \geq \frac{(\epsilon/8W_\epsilon^2 - l\sigma c_1(W_\epsilon^2 + 1))^2}{c_1^2 W_\epsilon^2 l} - 3\sigma l c_1(W_\epsilon^2 + 1) \tag{E.9}$$

In the second case $\mathbb{E}_{\theta \sim \rho_{t'}}[L'[\rho_{t'}](\theta)] \geq \frac{\epsilon}{2}$, $\forall t' \in [t, t+l]$. Therefore, we can integrate equation E.4 from $t$ to $t+l$ in order to get

$$L[\rho_t] - L[\rho_{t+l}] \geq l\frac{\epsilon^2}{4W_\epsilon^2} - \sigma l B_L(W_\epsilon^2 + 1)$$

Therefore, applying Lemma E.13 again gives

$$L[\rho_t] - L[\rho_{t+2l}] \geq l\frac{\epsilon^2}{4W_\epsilon^2} - 2\sigma l B_L(W_\epsilon^2 + 1) \tag{E.10}$$

Thus equation E.8 follows.

Now recall that we choose

$$\sigma \triangleq \exp(-d\log(1/\epsilon)\mathrm{poly}(k, M_V, M_R, M_\Phi, b_v, B_V, C_R, B_\Phi, L[\rho_0] - L[\rho^\star]))$$

For the simplicity, in the remaining computation, we will use $O(\cdot)$ notation to hide polynomials in the problem parameters besides $d, \epsilon$. We simply write $\sigma = \exp(-c_3 d\log(1/\epsilon))$. Recall our choice $t_\epsilon \triangleq O(\frac{d^2}{\epsilon^4}\log^2(1/\epsilon))$. It suffices to show that our objective would have sufficiently decreased in $t_\epsilon$ steps. We first note that with $c_3$ sufficiently large, $W_\epsilon^2 = O(L[\rho_0]/b_v) = O(1)$. Simplifying our expression for $l$, we get that $l = O(\frac{d}{\epsilon}\log\frac{1}{\epsilon})$, so long as $\sigma W_\epsilon^2 = o(\epsilon)$, which holds for sufficiently large $c_3$. Now let

$$\delta_1 \triangleq \frac{(\epsilon/8W_\epsilon^2 - l\sigma c_1(W_\epsilon^2 + 1))^2}{c_1^2 W_\epsilon^2 l} - 3\sigma l c_1(W_\epsilon^2 + 1)$$

$$\delta_2 \triangleq l\frac{\epsilon^2}{4W_\epsilon^2} - 2\sigma l B_L(W_\epsilon^2 + 1)$$

Again, for sufficiently large $c_3$, the terms with $\sigma$ become negligible, and $\delta_1 = O(\frac{\epsilon^2}{l}) = O(\frac{\epsilon^3}{d\log(1/\epsilon)})$. Likewise, $\delta_2 = O(d\epsilon\log(1/\epsilon))$.

Thus, if by time $t$ we have not encountered $2\epsilon$-optimal $\rho_t$, then we will decrease the objective by $O(\frac{\epsilon^3}{d\log(1/\epsilon)})$ in $O(\frac{d}{\epsilon}\log\frac{1}{\epsilon})$ time. Therefore, a total of $O(\frac{d^2}{\epsilon^4}\log^2(1/\epsilon))$ time is sufficient to obtain $\epsilon$ accuracy. $\qquad\square$

In the following section, we will complete the proofs of Lemmas E.5, E.6, E.7, and E.8.

### E.3  Missing Proofs for Theorem E.4

In this section, we complete the proofs of Lemmas E.5, E.6, E.7, and E.8. We first collect some general lemmas which will be useful in these proofs. The following general lemma computes integrals over vector field divergences.

**Lemma E.9.** *For any $h_1 : \mathbb{R}^{d+1} \to \mathbb{R}$, $h_2 : \mathbb{R}^{d+1} \to \mathbb{R}^{d+1}$ and distribution $\rho$ with $\rho(\theta) \to 0$ as $\|\theta\| \to \infty$,*

$$\int h_1(\theta)\nabla \cdot (h_2(\theta)\rho(\theta))d\theta = -E_{\theta\sim\rho}[\langle\nabla h_1(\theta), h_2(\theta)\rangle]$$

*Proof.* The proof follows from integration by parts. $\qquad\square$

We note that $\rho_t$ will satisfy the boundedness condition of Lemma E.9 during the course of our algorithm - $\rho_0$ starts with this property, and Lemma E.5 proves that $\rho_t$ will continue to have this property. We therefore freely apply Lemma E.9 in the remaining proofs. Now we bound the absolute value of $L'[\rho_t]$ over the sphere by $B_L$.

**Lemma E.10.** *For any $\bar{\theta} \in \mathbb{S}^{d-1}, t \geq 0$, $|L'[\rho_t](\bar{\theta})| \leq B_L$.*

*Proof.* We compute

$$|L'[\rho_t](\bar{\theta})| = \left|\left\langle\nabla R\left(\int \Phi d\rho\right), \Phi(\bar{\theta})\right\rangle + V(\bar{\theta})\right|$$

$$\leq \left\|\nabla R\left(\int \Phi d\rho\right)\right\|_2 \|\Phi(\bar{\theta})\|_2 + V(\bar{\theta}) \leq M_R B_\Phi + B_V$$

$\qquad\square$

The next lemma analyzes the decrease in $L[\rho_t]$ due to the gradient flow dynamics.

**Lemma E.11.** *Under the perturbed Wasserstein gradient flow*

$$\frac{d}{dt}L[\rho_t] = -\sigma\mathbb{E}_{\theta\sim\rho_t}[L'[\rho_t](\theta)] + \sigma\mathbb{E}_{\bar{\theta}\sim U^d}[L'[\rho_t](\bar{\theta})] - \mathbb{E}_{\theta\sim\rho_t}[\|v[\rho_t](\theta)\|_2^2]$$

*Proof.* Applying the chain rule, we can compute

$$\frac{d}{dt}L[\rho_t] = \left\langle \nabla R\left(\int \Phi d\rho_t\right), \frac{d}{dt}\int \Phi d\rho_t\right\rangle + \frac{d}{dt}\int V d\rho_t$$

$$= \frac{d}{dt}\mathbb{E}_{\theta \sim \rho_t}[L'[\rho_t](\theta)]$$

$$= \int L'[\rho_t](\theta)\rho_t'(\theta)d\theta$$

$$= -\sigma\int L'[\rho_t]d\rho_t + \sigma\int L'[\rho_t]dU^d - \int L'[\rho_t](\theta)\nabla\cdot(v[\rho_t](\theta)\rho_t(\theta))d\theta$$

$$= -\sigma\mathbb{E}_{\theta \sim \rho_t}[L'[\rho_t](\theta)] + \sigma\mathbb{E}_{\bar{\theta} \sim U^d}[L'[\rho_t](\bar{\theta})] - \mathbb{E}_{\theta \sim \rho_t}[\|v[\rho_t](\theta)\|_2^2],$$

where we use Lemma E.9 with $h_1 = L'[\rho_t]$ and $h_2 = v[\rho_t]$. $\qquad\square$

By combining the above Lemma with Lemma E.10, it follows that at the decrease in objective value is approximately the average velocity of all parameters under $\rho_t$ plus some additional noise on the scale of $\sigma$. At the end, we choose $\sigma$ small enough so that the noise terms essentially do not matter.

**Corollary E.12.** *We can bound $\frac{d}{dt}L[\rho_t]$ by*

$$\frac{d}{dt}L[\rho_t] \leq \sigma B_L(W_t^2 + 1) - \mathbb{E}_{\theta \sim \rho_t}[\|v[\rho_t](\theta)\|_2^2] \qquad (\text{E.11})$$

*Proof.* By homogeneity, and Lemma E.10, $\mathbb{E}_{\theta \sim \rho_t}[L'[\rho_t](\theta)] = \mathbb{E}_{\theta \sim \rho_t}[L'[\rho_t](\bar{\theta})\|\theta\|_2^2] \leq B_L W_t^2$. We also get $\mathbb{E}_{\bar{\theta} \sim U^d}[L'[\rho_t](\bar{\theta})] \leq B_L$ since $U^d$ is only supported on $\mathbb{S}^d$. Combining these with Lemma E.11 gives the desired statement. $\qquad\square$

Corollary E.12 implies that if we run the dynamics for a short time, the second moment of $\rho_t$ will grow slowly, again at a rate that is roughly the scale of the noise $\sigma$. This allows us to complete the proof of Lemma E.5.

*Proof of Lemma E.5.* Let $t^* \triangleq \arg\max_{t' \in [0,t]} W_{t'}^2$. Integrating both sides of equation E.11, and rearranging, we get

$$0 \leq \int_0^{t^*} \mathbb{E}_{\theta \sim \rho_s}[\|v[\rho_s](\theta)\|_2^2]ds \leq L[\rho_0] - L[\rho_t] + \sigma B_L\int_0^{t^*}(W_s^2 + 1)ds$$

$$\leq L[\rho_0] - L[\rho_{t^*}] + t^*\sigma B_L(W_{t^*}^2 + 1)$$

Now since $R$ is nonnegative, we apply $L[\rho_{t^*}] \geq E_{\theta \sim \rho_{t^*}}[V(\theta)] \geq E_{\theta \sim \rho_{t^*}}[V(\bar{\theta})\|\theta\|_2^2] \geq b_V W_{t^*}^2$. We now plug this in and rearrange to get $W_{t'}^2 \leq W_{t^*}^2 \leq \frac{L[\rho_0] + t^*\sigma B_L}{b_V - t^*\sigma B_L} \leq \frac{L[\rho_0] + t\sigma B_L}{b_V - t\sigma B_L} \forall 0 \leq t' \leq t$.

From the proof above, it immediately follows that $\forall 0 \leq t \leq t_\epsilon, W_t^2 \leq W_\epsilon^2$. $\qquad\square$

The next statement allows us to argue that our dynamics will never increase the objective by too much.

**Lemma E.13.** *For any $t_1, t_2$ with $0 \leq t_1 \leq t_2 \leq t_\epsilon$, $L[\rho_{t_2}] - L[\rho_{t_1}] \leq \sigma(t_2 - t_1)B_L(W_\epsilon^2 + 1)$.*

*Proof.* From Corollary E.12, $\forall t \in [t_1, t_2]$ we have

$$\frac{d}{dt}L[\rho_t] \leq \sigma B_L(W_\epsilon^2 + 1)$$

Integrating from $t_1$ to $t_2$ gives the desired result. $\qquad\square$

The following lemma bounds the change in expectation of a 2-homogeneous function over $\rho_t$. At a high level, we lower bound the decrease in our loss as a function of the change in this expectation. By applying this lemma, we will be able to prove Lemma E.6.

**Lemma E.14.** *Let* $h : \mathbb{R}^{d+1} \to \mathbb{R}$ *that is 2-homogeneous, with* $\|\nabla h(\bar{\theta})\| \leq M \ \forall \bar{\theta} \in \mathbb{S}^d$ *and* $|h(\bar{\theta})| \leq B \ \forall \bar{\theta} \in \mathbb{S}^d$. *Then* $\forall 0 \leq t \leq t_\epsilon$, *we have*

$$\left| \frac{d}{dt} \int h d\rho_t \right| \leq \sigma B(W_\epsilon^2 + 1) + MW_\epsilon \left( -\frac{d}{dt} L[\rho_t] + \sigma B_L(W_\epsilon^2 + 1) \right)^{1/2} \quad \text{(E.12)}$$

*Proof.* Let $Q(t) \triangleq \int h d\rho_t$. We can compute:

$$\begin{aligned}
Q'(t) &= \int h(\theta) \frac{d\rho_t}{dt}(\theta) d\theta \\
&= \int h(\theta)(-\sigma\rho_t(\theta) - \nabla \cdot (v[\rho_t](\theta)\rho_t(\theta))) d\theta + \sigma \int h dU^d \\
&= -\sigma \int h(\bar{\theta})\|\theta\|_2^2 \rho_t(\theta) d\theta + \sigma \int h dU^d - \int h(\theta) \nabla \cdot (v[\rho_t](\theta)\rho_t(\theta)) d\theta \quad \text{(E.13)}
\end{aligned}$$

Note that the first two terms are bounded by $\sigma B(W_\epsilon^2 + 1)$ by the assumptions for the lemma. For the third term, we have from Lemma E.9:

$$\begin{aligned}
\left| \int h(\theta) \nabla \cdot (v[\rho_t](\theta)\rho_t(\theta)) d\theta \right| &= |E_{\theta \sim \rho_t}[\langle \nabla h(\theta), v[\rho_t](\theta) \rangle]| \\
&\leq \sqrt{E_{\theta \sim \rho_t}[\|\nabla h(\theta)\|_2^2] E_{\theta \sim \rho_t}[\|v[\rho_t](\theta)\|_2^2]} \quad \text{(by Cauchy-Schwarz)} \\
&\leq \sqrt{E_{\theta \sim \rho_t}[\|\nabla h(\bar{\theta})\|_2^2 \|\theta\|_2^2] E_{\theta \sim \rho_t}[\|v[\rho_t](\theta)\|_2^2]} \quad \text{(by homogeneity of } \nabla h) \\
&\leq MW_\epsilon \sqrt{E_{\theta \sim \rho_t}[\|v[\rho_t](\theta)\|_2^2]} \quad \text{(since } h \text{ is Lipschitz on the sphere)} \\
&\leq MW_\epsilon \left( -\frac{d}{dt} L[\rho_t] + \sigma B_L(W_\epsilon^2 + 1) \right)^{1/2} \quad \text{(by Corollary E.12)}
\end{aligned}$$

Plugging this into equation E.13, we get that

$$|Q'(t)| \leq \sigma B(W_\epsilon^2 + 1) + MW_\epsilon \left( -\frac{d}{dt} L[\rho_t] + \sigma B_L(W_\epsilon^2 + 1) \right)^{1/2}$$

$\square$

Now we complete the proof of Lemma E.6.

*Proof of Lemma E.6.* Recall that $L'[\rho_t](\bar{\theta}) = \langle \nabla R(\int \Phi d\rho_t), \Phi(\bar{\theta}) \rangle + V(\bar{\theta})$. Differentiating with respect to $t$,

$$\begin{aligned}
\frac{d}{dt} L'[\rho_t](\bar{\theta}) &= \left\langle \frac{d}{dt} \nabla R \left( \int \Phi d\rho_t \right), \Phi(\bar{\theta}) \right\rangle \\
&= \Phi(\bar{\theta})^\top \nabla^2 R(Q(t)) Q'(t) \\
&\leq C_R B_\Phi \|Q'(t)\|_2 \\
&\leq C_R B_\Phi \|Q'(t)\|_1 \quad \text{(E.14)}
\end{aligned}$$

Integrating and applying the same reasoning to $-L'[\rho_t]$ gives us equation E.2. Now we apply Lemma E.14 to get

$$\begin{aligned}
\|Q'(t)\|_1 &= \sum_{i=1}^k \left| \frac{d}{dt} \int \Phi_i d\rho_t \right| \\
&\leq \sum_{i=1}^k \left[ \sigma B_\Phi(W_\epsilon^2 + 1) + M_\Phi W_\epsilon \left( -\frac{d}{dt} L[\rho_t] + \sigma B_L(W_\epsilon^2 + 1) \right)^{1/2} \right] \\
&\leq k\sigma B_\Phi(W_\epsilon^2 + 1) + k M_\Phi W_\epsilon \left( -\frac{d}{dt} L[\rho_t] + \sigma B_L(W_\epsilon^2 + 1) \right)^{1/2}
\end{aligned}$$

We plug this into equation E.14 and then integrate both sides to obtain

$$C_R B_\Phi \int_t^{t+l} \|Q'(t)\|_1$$

$$\leq k\sigma l C_R B_\Phi^2 (W_\epsilon^2 + 1) + k C_R B_\Phi M_\Phi W_\epsilon \int_t^{t+l} \left( -\frac{d}{dt} L[\rho_t] + \sigma B_L (W_\epsilon^2 + 1) \right)^{1/2}$$

$$\leq k\sigma l C_R B_\Phi^2 (W_\epsilon^2 + 1) + k C_R B_\Phi M_\Phi W_\epsilon \sqrt{l} (L[\rho_t] - L[\rho_{t+l}] + \sigma l B_L (W_\epsilon^2 + 1))^{1/2}$$

Using $c_1 \triangleq \max\{k C_R B_\Phi^2, k C_R B_\Phi M_\Phi, B_L\}$ gives the statement in the lemma. $\qquad \square$

Now we will fill in the proof of Lemma E.8. We first show that $L'$ is Lipschitz on the unit ball. Recall that in the statement of Lemma E.8, we define a constant $c_2$ by $c_2 \triangleq \sqrt{k} M_R M_\Phi + M_V$.

**Lemma E.15.** *For all $\bar{\theta}, \bar{\theta}' \in \mathbb{S}^d$,*

$$|L'[\rho](\bar{\theta}) - L'[\rho](\bar{\theta}')| \leq c_2 \|\bar{\theta} - \bar{\theta}'\|_2 \tag{E.15}$$

*Proof.* Using the definition of $L'$ and triangle inequality,

$$|L'[\rho](\bar{\theta}) - L'[\rho](\bar{\theta}')| \leq \left\| \nabla R \left( \int \Phi d\rho \right) \right\|_2 \|\Phi(\bar{\theta}) - \Phi(\bar{\theta}')\|_2 + |V(\bar{\theta}) - V(\bar{\theta}')|$$

$$\leq (\sqrt{k} M_R M_\Phi + M_V)\|\bar{\theta} - \bar{\theta}'\|_2 \qquad \text{(by definition of } M_\Phi, M_R, M_V)$$

$\qquad\qquad\qquad\qquad\qquad\qquad\qquad\qquad\qquad\qquad\qquad\qquad\qquad\qquad\square$

Next, we introduce notation to refer to the $-\tau$-sublevel set of $L'[\rho_t]$. This will be useful for our proof of Lemma E.8. Define $K_t^{-\tau} \triangleq \{\bar{\theta} \in \mathbb{S}^d : L'[\rho_t](\bar{\theta}) \leq -\tau\}$, the $-\tau$-sublevel set of $L'[\rho_t]$, and let $m(S) \triangleq \mathbb{E}_{\theta \sim U^d}[\mathbf{1}(\theta \in S)]$ be the normalized spherical area of the set $S$. The following statement uses the Lipschitz-ness of $L'[\rho_t]$ to lower bound the volume of $K_t^{-\tau+\delta}$ for some $\delta > 0$ if the $-\tau$-sublevel set $K_t^{-\tau}$ is nonempty.

**Lemma E.16.** *If $K_t^{-\tau}$ is nonempty, for $0 \leq \delta \leq \tau$, $\log m(K_t^{-\tau+\delta}) \geq -2d \log \frac{c_2}{\delta}$.*

*Proof.* Let $\bar{\theta} \in K_t^{-\tau}$. From Lemma E.15, $L'[\rho](\bar{\theta}') \leq -\tau + \delta$ for all $\bar{\theta}'$ with $\|\bar{\theta}' - \bar{\theta}\|_2 \leq \frac{\delta}{c_2}$. Thus, we have

$$m(K_t^{-\tau+\delta}) \geq \mathbb{E}_{\bar{\theta}' \sim U^d} \left[ \mathbf{1}[\|\bar{\theta}' - \bar{\theta}\|_2 \leq \frac{\delta}{c_2}] \right]$$

Now the statement follows by Lemma 2.3 of [8]. $\qquad\qquad\qquad\qquad\qquad\qquad\qquad\qquad\square$

Finally, the proof of Lemma E.8 will require a general lemma about the magnitude of the gradient of a 2-homogeneous function in the radial direction.

**Lemma E.17.** *Let $h : \mathbb{R}^{d+1} \to \mathbb{R}$ be a 2-homogeneous function. Then for any $\theta \in \mathbb{R}^{d+1}$, $\theta^\top \nabla h(\theta) = 2\|\theta\|_2 h(\bar{\theta})$.*

*Proof.* We have $h(\theta + \alpha\bar{\theta}) = (\|\theta\|_2 + \alpha)^2 h(\bar{\theta})$. Differentiating both sides with respect to $\alpha$ and evaluating the derivative at 0, we get $\theta^\top \nabla h(\theta) = 2\|\theta\|_2 h(\bar{\theta})$, as desired. $\qquad\square$

Now we are ready to complete the proof of Lemma E.8. Recall that in the setting of Lemma E.8, $l$ is the length of the time interval over which the descent direction causes a decrease in the objective. We will first show that a descent direction in $L'[\rho_t]$ will remain for the next $l$ time steps. In the notation of Lemma E.6, define $z(s) \triangleq C_R B_\Phi \int_{t^*}^{t^*+s} \|Q'(t)\|_1 dt$. Note that from Lemma E.6, for all $\bar{\theta} \in \mathbb{S}^d$ we have $|L'[\rho_{t^*+s}](\bar{\theta}) - L'[\rho_{t^*}](\bar{\theta})| \leq z(s)$. Thus, the following holds:

**Claim E.18.** *For all $s \leq l$, $K_{t^*+s}^{-\tau+z(s)}$ is nonempty.*

*Proof.* By assumption, $\exists \bar{\theta}$ with $\bar{\theta} \in K_{t^*}^{-\tau}$. Then $L'[\rho_{t^*+s}](\bar{\theta}) \leq L'[\rho_{t^*}](\bar{\theta}) + z(s) \leq -\tau + z(s)$, so $K_{t^*+s}^{-\tau+z(s)}$ is nonempty. $\qquad \square$

Let $T_s \triangleq K_{t^*+s}^{-\tau/2+z(s)}$ for $0 \leq s \leq l$. We now argue that this set $T_s$ does not shrink as $t$ increases.

**Claim E.19.** *For all $s' > s$, $T_{s'} \supseteq T_s$.*

*Proof.* From equation E.14 and the definition of $z(s)$, $|L'[\rho_{t+s'}](\bar{\theta}) - L'[\rho_{t+s}](\bar{\theta})| \leq z(s') - z(s)$. It follows that for $\bar{\theta} \in T_s$

$$
\begin{aligned}
L'[\rho_{t+s'}](\bar{\theta}) &\leq L'[\rho_{t+s}](\bar{\theta}) + z(s') - z(s) \\
&\leq -\tau/2 + z(s) - z(s) + z(s') &\text{(by definition of } T_s) \\
&\leq -\tau/2 + z(s')
\end{aligned}
$$

which means that $\bar{\theta} \in T_{s'}$. $\qquad \square$

Now we show that the weight of the particles in $T_s$ grows very fast if $z(k)$ is small.

**Claim E.20.** *Suppose that $z(l) \leq \tau/4$. Let $\tilde{T}_s = \{\theta \in \mathbb{R}^{d+1} : \bar{\theta} \in T_s\}$. Define $N(s) \triangleq \int_{\tilde{T}_s} \|\theta\|^2 d\rho_{t^*+s}$ and $\beta \triangleq \exp(-2d \log \frac{2c_2}{\tau})$. Then $N'(s) \geq (\tau - \sigma)N(s) + \sigma\beta$.*

*Proof.* From the assumption $z(l) \leq \frac{\tau}{4}$, it holds that $T_s \subseteq K_{t^*+s}^{-\tau/4} \ \forall s \leq k$. Since $T_s$ is defined as a sublevel set, $v[\rho_{t^*+s}](\bar{\theta})$ points inwards on the boundary of $T_s$ for all $\bar{\theta} \in T_s$, and by 1-homogeneity of the gradient, the same must hold for all $u \in \tilde{T}_s$.

Now consider any particle $\theta \in \tilde{T}_s$. We have that $\theta$ flows to $\theta + v[\rho_{t^*+s}](\theta)ds$ at time $t^* + s + ds$. Furthermore, since the gradient points inwards from the boundary, it also follows that $u + v[\rho_{t^*+s}](\theta)ds \in \tilde{T}_s$. Now we compute

$$
\begin{aligned}
\int_{\tilde{T}_s} \|\theta\|_2^2 d\rho_{t^*+s+ds} &= (1-\sigma ds) \int_{\tilde{T}_s} \|\theta + v[\rho_{t^*+s}](\theta)ds\|_2^2 d\rho_{t^*+s} + \sigma ds \int_{\tilde{T}_s} 1 dU^d \\
&\geq (1-\sigma ds) \int_{\tilde{T}_s} (\|\theta\|_2^2 + 2\theta^\top v[\rho_{t^*+s}](\theta)ds) d\rho_{t^*+s} + \sigma m(K_{t^*+s}^{-\tau/2+z(s)})ds
\end{aligned}
\tag{E.16}
$$

Now we apply Lemma E.17, using the 2-homogeneity of $F'$ and the fact that $L'[\rho_{t^*+s}](\bar{\theta}) \leq -\tau/4 \ \forall \theta \in \tilde{T}_s$

$$
\begin{aligned}
\|\theta\|_2^2 + 2\theta^\top v[\rho_{t^*+s}](\theta)ds = \|\theta\|_2^2 - 4\|\theta\|_2^2 L'[\rho_{t^*+s}](\bar{\theta})ds \\
\geq \|\theta\|_2^2(1 + \tau ds)
\end{aligned}
\tag{E.17}
$$

Furthermore, since $K_{t^*+s}^{-\tau+z(s)}$ is nonempty by Claim E.18, we can apply Lemma E.16 and obtain

$$
m(K_{t^*+s}^{-\tau/2+z(s)}) \geq \beta
\tag{E.18}
$$

Plugging equation E.17 and equation E.18 back into equation E.16, we get

$$
\int_{\tilde{T}_s} \|u\|_2^2 d\rho_{t^*+s+ds} \geq (1-\sigma ds)(1 + 2\tau ds)N(s) + \sigma\beta ds
$$

Since we also have that $\tilde{T}_{s+ds} \supseteq \tilde{T}_s$, it follows that

$$
N(s+ds) = \int_{\tilde{T}_{s+ds}} \|u\|_2^2 d\rho_{t^*+s+ds} \geq (1-\sigma ds)(1 + \tau ds)N(s) + \sigma\beta ds
$$

and so $N'(s) \geq (\tau - \sigma)N(s) + \sigma\beta$. $\qquad \square$

Using Claim E.20 allows us to complete the proof of Lemma E.8.

*Proof of Lemma E.8.* If $z(l) = C_R B_\Phi \int_t^{t+l} \|Q'(t)\|_1 \geq \frac{\tau}{4}$, then by rearranging the conclusion of Lemma E.6 we immediately get equation E.5.

Suppose for the sake of contradiction that $z(l) \leq \tau/4$. From Claim E.20, it follows that $N(1) \geq \sigma\beta$, and $N(l) \geq \exp((\tau-\sigma)(l-1))N(1)$. Thus, in $\frac{\log(W_\epsilon^2/\sigma) + 2d\log\frac{2c_2}{\tau}}{\tau-\sigma} + 1$ time, $W_{t^*+l} \geq N(l) \geq W_\epsilon^2$, a contradiction. Therefore, it must be true that $z(l) \geq \tau/4$.

$\square$

Finally, we fill in the proof of Lemma E.7.

*Proof.* We can first compute

$$\mathbb{E}_{\theta\sim\rho_t}[L'[\rho_t](\theta)] = \mathbb{E}_{\theta\sim\rho_t}[L'[\rho_t](\bar{\theta})\|\theta\|_2^2]$$

$$= \frac{1}{2}\mathbb{E}_{\theta\sim\rho_t}[\|\theta\|_2\bar{\theta}^\top v[\rho_t](\theta)] \qquad\qquad \text{(via Lemma E.17)}$$

$$\leq \frac{1}{2}\sqrt{\mathbb{E}_{\theta\sim\rho_t}[\|\theta\|_2^2]\mathbb{E}_{\theta\sim\rho_t}[\|v[\rho_t](\theta)\|_2^2]} \qquad\qquad \text{(by Cauchy-Schwarz)}$$

$$\leq \frac{1}{2}W_\epsilon\sqrt{\mathbb{E}_{\theta\sim\rho_t}[\|v[\rho_t](\theta)\|_2^2]}$$

Rearranging gives $\mathbb{E}_{\theta\sim\rho_t}[\|v[\rho_t](\theta)\|_2^2] \geq \frac{\mathbb{E}_{\theta\sim\rho_t}[L'[\rho_t](\theta)]^2}{W_\epsilon^2}$, and plugging this into equation E.11 gives the desired result. $\square$

## E.4 Discrete-Time Optimization

To circumvent the technical issue of existence of a solution to the continuous-time dynamics, we also note that polynomial time convergence holds for discrete-time updates.

**Theorem E.21.** *Along with Assumptions E.1, E.2, E.3 additionally assume that $\nabla\Phi_i$ and $\nabla V$ are $C_\Phi$ and $C_V$-Lipschitz, respectively. Let $\rho_t$ evolve according to the following discrete-time update:*

$$\rho_{t+1} \triangleq \rho_t + \eta(-\sigma\rho_t + \sigma U^d - \nabla \cdot (v[\rho_t]\rho_t))$$

*There exists a choice of*

$$\sigma \triangleq \exp(-d\log(1/\epsilon)\text{poly}(k, M_V, M_R, b_V, B_V, C_R, B_\Phi, C_\Phi, C_V, L[\rho_0] - L[\rho^\star]))$$

$$\eta \triangleq \text{poly}(k, M_V, M_R, b_V, B_V, C_R, B_\Phi, C_\Phi, C_V, L[\rho_0] - L[\rho^\star])$$

$$t_\epsilon \triangleq \frac{d^2}{\epsilon^4}\text{poly}(k, M_V, M_R, b_V, B_V, C_R, B_\Phi, C_\Phi, C_V, L[\rho_0] - L[\rho^\star])$$

*such that $\min_{0\leq t\leq t_\epsilon} L[\rho_t] - L^\star \leq \epsilon$.*

The proof follows from a standard conversion of the continuous-time proof of Theorem E.4 to discrete time, and we omit it here for simplicity.

# F Additional Simulations

In this section we provide more details on the simulations described in Section 5. The experiments were small enough to run on a standard computer, though we used a single NVIDIA TitanXp GPU. We decided the value of regularization $\lambda$ based on the training length - longer training time meant we could use smaller $\lambda$.

## F.1 Test Error and Margin vs. Hidden Layer Size

To justify Theorem 4.3, we also plot the dependence of the test error and margin on the hidden layer size in Figure 3 for synthetic data generated from a ground truth network with 10 hidden units and also MNIST. The plots indicate that test error is decreasing in hidden layer size while margin is increasing, as Theorem 4.3 predicts. We train the networks for a long time in this experiment: we train for 80000 passes on the synthetic data and 600 epochs for MNIST.

Figure 3: Dependence of margin and test error on hidden layer size. **Left:** Synthetic. **Right:** MNIST.

Figure 4: Neural nets vs. kernel method with $r_w = 0, r_u = 1$ (Theorem 2.1 setting). **Left:** Classification. **Right:** Regression.

The left side of Figure 3 shows the experimental results for synthetic data generated from a ground truth network with 10 hidden units, input dimension $d = 20$, and a ground truth unnormalized margin of at least 0.01. We train for 80000 steps with learning rate 0.1 and $\lambda = 10^{-5}$, using two-layer networks with $2^i$ hidden units for $i$ ranging from 4 to 10. We perform 20 trials per hidden layer size and plot the average over trials where the training error hit 0. (At a hidden layer size of $2^7$ or greater, all trials fit the training data perfectly.) The right side of Figure 3 demonstrates the same experiment, but performed on MNIST with hidden layer sizes of $2^i$ for $i$ ranging from 6 to 15. We train for 600 epochs using a learning rate of 0.01 and $\lambda = 10^{-6}$ and use a single trial per plot point. For MNIST, all trials fit the training data perfectly. The MNIST experiments are more noisy because we run one trial per plot point for MNIST, but the same trend of decreasing test error and increasing margin still holds.

## F.2 Neural Net and Kernel Generalization vs. Training Set Size

We compare the generalization of neural nets and kernel methods for classification and regression. In Figure 4 we plot the generalization error of a trained neural net against a $\ell_2$ kernel method with relu features (corresponding to $r_1 = 0, r_2 = 1$ in the setting of Theorem 2.1) as we vary $n$. Our ground truth comes from a random neural network with 6 hidden units, and during training we use a network with as many hidden units as examples. For classification, we used rejection sampling to obtain datapoints with unnormalized margin of at least 0.1 on the ground truth network. We use a fixed dimension of $d = 20$. For all experiments, we train the network for 20000 steps with $\lambda = 10^{-8}$ and average over 100 trials for each plot point.

For classification we plot 0-1 error, whereas for regression we plot squared error. The plots show that two-layer nets clearly outperform the kernel method in test error as $n$ grows.

## F.3 Verifying Convergence to the Max-Margin

We verify the normalized margin convergence on a two-layer networks with one-dimensional input. A single hidden unit computes the following: $x \mapsto a_j \mathrm{relu}(w_j x + b_j)$. We add $\| \cdot \|_2^2$-regularization to $a, w$, and $b$ and compare the resulting normalized margin to that of an approximate solution of the $\ell_1$ SVM problem with features $\mathrm{relu}(wx_i + b)$ for $w^2 + b^2 = 1$. Writing this feature vector is intractable, so we solve an approximate version by choosing 1000 evenly spaced values of $(w, b)$. Our theory predicts that with decreasing regularization, the margin of the neural network converges to the $\ell_1$ SVM objective. In Figure 5, we plot this margin convergence and visualize the final networks and

Figure 5: Neural network with input dimension 1. **Left:** Normalized margin as we decrease $\lambda$. **Right:** Visualization of the normalized functions computed by the neural network and $\ell_1$ SVM solution for $\lambda \approx 2^{-14}$.

ground truth labels. The network margin approaches the ideal one as $\lambda \to 0$, and the visualization shows that the network and $\ell_1$ SVM functions are extremely similar.