[Reviews · NeurIPS 2019]

Reviewer 1



Summary: The paper studies the generalization and optimization aspects of regularized neural networks, and provide two key contributions: (a)they show that a O(d) sample complexity gap between global minima of regularized loss and the induced kernel method. (b). They also establish that in infinite-width two-layer nets, a variant of gradient descent converges to global minimum with of (weakly) regularized cross entropy loss in poly iterations. Detailed comments: 1. The paper studies a natural and important problem and makes fundamental contributions in this direction. Recent results in deep learning theory exploits this neural tangent connection to prove optimization and generalization results. In light of this, it is important to study the limitations of this. This highlights that recently popular neural tangent kernel view has its own drawbacks and perhaps(?) insufficient to explain the success of modern machine learning. 2. The paper is very well-written with the problem setting and intuitions clearly explained. In particular, the authors explain the intuition behind their lower bound: why the induced kernel method will require more samples. 3. The technical content is solid and rigorous, with novel contributions of tools which could be useful in general. Moreover, the high level ideas are well succinctly explained, for example, the key idea they exploit for the upper bound is showing that global minimizer of regularized loss converges to max margin solutions. Another important technical contribution they remark is a new technique for proving lower bound of kernel methods which is explained done exploiting the fact that the predictor in the RKHS lies in the span of the data. I skimmed through the proofs of generalization, and the arguments seemed fine and well written to me. 4. A good thing is that the authors remark the generality of the result presented; for example, the sample complexity gap is showed to hold for multi-class classification and regression. This answers natural questions on how general are the ideas to extend. The authors also remark these results can potentially hold even when the only regularization is the implicit regularization of optimization. 5. The only section that required multiple re-reads for me is Section 3. Perturbed Wassertien Flow. Perhaps this is because I am not familiar with relevant literature, but I had a feeling that it was rushed and should have been explained/presented better. For example, there is no intuition provided how eqn 3.1 came about or what it means. Similarly, for eqn 3.2, the authors explain what U is but what about the first term? A few more lines explaining this will help readability. This is important because it provides more accessibility to a broad audience as NeurIPS. Moreover, the authors don't give intuition/details of proof techniques for the optimization result in the main paper. I hope they polish section 3 in the revision. Minor comment: Please explain somewhere why it is called "weak" regularization.

Reviewer 2



This paper shows that regularization is helpful for the sample complexity in some settings. Specifically, they consider two-layer ReLU networks, and carefully construct a data distribution such that the optimal regularized neural net learns with O(d) samples while the NTK network requires \Omega(d^2) samples. Basically, their data distribution has very sparse features such that the regularization helps to find informative features while NTK overfits to noise. The authors also show that for infinite-width two-layer networks, noisy gradient descent can find global minimizers of the l_2 regularized function within polynomial time. For the proof, in order to upper bound the sample complexity for regularized neural nets, the authors show that with a small regularizer (goes to zero), the global minimizer of regularized logistic loss will converge to a maximum margin solution. This holds for any positive homogeneous predictor, including multi-layer ReLU nets. The authors develop new techniques to lower bound the sample complexity of NTK. These techniques can be applied to other cases. Overall, this is a good theory paper addressing important problems with novel techniques. The paper is well written. Here are my main comments: 1. The optimization result of the infinite-width nets with regularized loss function requires the activation function to be differentiable, which does not hold for ReLU. I was wondering whether some smoothing techniques can resolve this issue. It might be better to discuss the challenges to extend to ReLU nets and the possible approaches. 2. Theorem 4.3 only means using a wider network won’t hurt the generalization bound. It will be more interesting if the authors can show using a wider network can strictly improve the maximum margin achievable. Of course, this may not be true. The authors can discuss a bit more about the possibility of establishing a gap of generalizations between narrow nets and wide nets. Here are some minor comments: 1. It might be better to move Figure 1 to page 4 since the figure is an illustration of the distribution. 2. Line 44: than than’’ 3. Line 126: a ab’’ ------------------------------------------------------------------ I thank the authors' clarifications, which have successfully addressed my questions. I will keep my score as it is.

Reviewer 3



Originality: One of the novel contribution of this paper is the proof of a lower bound of the sample complexity of the NTK, which requires to explore the following observation mathematically: NTK could not focus on the informative signal of the inputs. Quality: I think this is a high quality paper, which contains a lot of contents. In my opinion, Theorem 2.1 alone together with some additional experiments would suffice to make an interesting paper. Clarity: The paper is very well-written. Thought it is quite technical, the authors make some effort in explaining the intuition and ideas behind some of the technical proofs. Significance: I think this is a good paper that helps us better understand the NTK and its limitation. -------------------------------------------------------------------------------------------- Update: I will keep my score.

[Author Response · NeurIPS 2019]

We thank the reviewers for the detailed and insightful reviews. As the reviewers noted, our work 1) contributes to "a
deeper understanding of NTK and its limitations" and 2) develops novel analysis tools and techniques. We answer
reviewers' questions below and will incorporate feedback into the final revision.

**Reviewer 1:**

—"there is no intuition provided how eqn 3.1 came about or what it means... explaining this will help readability."

Thank you for the valuable feedback on this section — we will incorporate this in our next revision. Equation 3.1 and the
first term of Equation 3.2 are due to the formula for the Wasserstein gradient flow dynamics (see, e.g., [Santambrogio,
2017]), which are derived via continuous time steepest descent with respect to Wasserstein distance over the space of
probability distributions on the neurons.

—"intuition/details of proof techniques for the optimization result in the main paper"

The intuition for the proof of Theorem 3.3 is that the optimization problem is convex over the space of probability
distributions on neurons. We use this convexity to argue that if there is a descent direction, the uniform noise (U in
equation 3.2) along with the 2-homogeneity will allow the optimization dynamics to increase the mass in this direction
exponentially fast, which results in decrease of the loss by a polynomial amount. Note that though the problem is convex
over the space of distributions, SGD on the network weights does not use the gradient with respect to the probability
distribution so the convergence claim is not immediate.

— "Minor comment: Please explain somewhere why it is called "weak" regularization."

By weak regularization, we refer to the fact that $\lambda \to 0$ for our Theorem 4.1 to hold.

**Reviewer 2:**

— "might be better to discuss the challenges to extend to ReLU nets and the possible approaches."

The difficulty with ReLU networks is that if the gradient flow pushes neurons towards 0, issues of differentiability arise.
One potential approach to circumvent this issue is arguing that with correct initialization, the iterates will never reach 0.

— "more interesting if the authors can show using a wider network can strictly improve the maximum margin achievable
... discuss a bit more . . . gap of generalizations between narrow nets and wide nets."

It is possible to construct an instance where a narrower and wider network can both fit the training data, but the wider
networks allow larger maximum normalized margin. Consider a distribution where the first two coordinates are the
same as in our construction for Theorem 2.1, and the third coordinate is $\pm\epsilon$ with sign matching the label. Then a two
neuron network separates this data with margin $\epsilon/2$ by using the third coordinate, whereas there exists a four neuron
network – the one described in line 166 of the paper – that separates this data with margin $1/4$. We suspect that an
intuition such as this can be used to prove a formal generalization gap (instead of only a gap of margins) based on width.
This is an interesting direction for future work and we thank the reviewer for this suggestion.

**Reviewer 3:**

— "prove a similar result of Theorem 2.1 when the first $k$ coordinates of the inputs are informative? And how the sample
complexity depends on $(k, d)$ as $k, d \to \infty$?"

It seems likely that some result in the form of Theorem 2.1 could be shown for some distribution where the first $k$ input
coordinates contain signal, although we have not fully explored this possibility at the moment. This is an interesting
direction for future work and we thank the reviewer for this suggestion.

— "how do the sample complexity in Thm 2.1 change if adding a regularizer to NTK (i.e. ridge regression)?"

Theorem 2.1 will still apply in this setting. The optimizer of ridge regression with NTK will lie in the RKHS spanned
by the data, and the sample complexity lower bound of Theorem 2.1 applies to all such functions.

— "possible to prove that for regularized *finite* width NN, SGD could learn distribution (2.1)?"

Empirically, we have found that SGD on finite width networks does indeed learn distribution (2.1). With over-
parameterization, each of the learned neurons converges to one of the directions $e_1, -e_1, e_2, -e_2$. The question of
rigorously proving that this behavior holds is an interesting and very challenging question for future work.

# References

Filippo Santambrogio. {Euclidean, metric, and Wasserstein} gradient flows: an overview. *Bulletin of Mathematical*
*Sciences*, 7(1):87–154, 2017.


[Meta-Review · NeurIPS 2019]

This paper investigates how the regularization helps for training neural networks in contrast to the unregularized neural tangent kernel method. It is shown that regularization captures "informative signal" but the NTK model does not, which highlights the effectiveness of the regularization. Moreover, this paper shows polynomial time convergence of gradient flow corresponding to the infinite width neural network. The contribution is novel and the implication is quite instructive to neural tangent kernel learning. Especially, the lower bound evaluation for kernel learning is a novel contribution.